# Hypothesis Set Stability and Generalization

**Dylan J. Foster**
Massachusetts Institute of Technology
dylanf@mit.edu

**Spencer Greenberg**
Spark Wave
admin@sparkwave.tech

**Satyen Kale**
Google Research
satyen@satyenkale.com

**Haipeng Luo**
University of Southern California
haipengl@usc.edu

**Mehryar Mohri**
Google Research and Courant Institute
mohri@google.com

**Karthik Sridharan**
Cornell University
sridharan@cs.cornell.edu

## Abstract

We present a study of generalization for data-dependent hypothesis sets. We give a general learning guarantee for data-dependent hypothesis sets based on a notion of transductive Rademacher complexity. Our main result is a generalization bound for data-dependent hypothesis sets expressed in terms of a notion of *hypothesis set stability* and a notion of Rademacher complexity for data-dependent hypothesis sets that we introduce. This bound admits as special cases both standard Rademacher complexity bounds and algorithm-dependent uniform stability bounds. We also illustrate the use of these learning bounds in the analysis of several scenarios.

## 1 Introduction

Most generalization bounds in learning theory hold for a fixed hypothesis set, selected before receiving a sample. This includes learning bounds based on covering numbers, VC-dimension, pseudo-dimension, Rademacher complexity, local Rademacher complexity, and other complexity measures [Pollard, 1984, Zhang, 2002, Vapnik, 1998, Koltchinskii and Panchenko, 2002, Bartlett et al., 2002]. Some alternative guarantees have also been derived for specific algorithms. Among them, the most general family is that of uniform stability bounds given by Bousquet and Elisseeff [2002]. These bounds were recently significantly improved by Feldman and Vondrak [2019], who proved guarantees that are informative, even when the stability parameter $\beta$ is only in $o(1)$, as opposed to $o(1/\sqrt{m})$. The $\log^2 m$ factor in these bounds was later reduced to $\log m$ by Bousquet et al. [2019]. Bounds for a restricted class of algorithms were also recently presented by Maurer [2017], under a number of assumptions on the smoothness of the loss function. Appendix A gives more background on stability.

In practice, machine learning engineers commonly resort to hypothesis sets depending on the *same sample* as the one used for training. This includes instances where a regularization, a feature transformation, or a data normalization is selected using the training sample, or other instances where the family of predictors is restricted to a smaller class based on the sample received. In other instances, as is common in deep learning, the data representation and the predictor are learned using the same sample. In ensemble learning, the sample used to train models sometimes coincides with the one used to determine their aggregation weights. However, standard generalization bounds are not directly applicable for these scenarios since they assume a fixed hypothesis set.

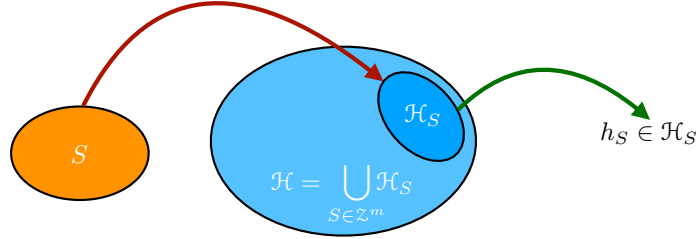

Figure 1: Decomposition of the learning algorithm's hypothesis selection into two stages. In the first stage, the algorithm determines a hypothesis $\mathcal{H}_S$ associated to the training sample $S$ which may be a small subset of the set of all hypotheses that could be considered, say $\mathcal{H} = \bigcup_{S \in \mathcal{Z}^m} \mathcal{H}_S$. The second stage then consists of selecting a hypothesis $h_S$ out of $\mathcal{H}_S$.

## 1.1 Contributions of this paper.

**1. Foundational definitions of data-dependent hypothesis sets.** We present foundational definitions of learning algorithms that rely on *data-dependent* hypothesis sets. Here, the algorithm decomposes into two stages: a first stage where the algorithm, on receiving the sample $S$, chooses a hypothesis set $\mathcal{H}_S$ dependent on $S$, and a second stage where a hypothesis $h_S$ is selected from $\mathcal{H}_S$. Standard generalization bounds correspond to the case where $\mathcal{H}_S$ is equal to some fixed $\mathcal{H}$ independent of $S$. Algorithm-dependent analyses, such as uniform stability bounds, coincide with the case where $\mathcal{H}_S$ is chosen to be a singleton $\mathcal{H}_S = \{h_S\}$. Thus, the scenario we study covers both existing settings and other intermediate scenarios. Figure 1 illustrates our general scenario.

**2. Learning bounds via transductive Rademacher complexity.** We present general learning bounds for data-dependent hypothesis sets using a notion of transductive Rademacher complexity (Section 3). These bounds hold for arbitrary bounded losses and improve upon previous guarantees given by Gat [2001] and Cannon et al. [2002] for the binary loss, which were expressed in terms of a notion of shattering coefficient adapted to the data-dependent case, and are more explicit than the guarantees presented by Philips [2005][corollary 4.6 or theorem 4.7]. Nevertheless, such bounds may often not be sufficiently informative, since they ignore the relationship between hypothesis sets based on similar samples.

**3. Learning bounds via hypothesis set stability.** We provide finer generalization bounds based on the key notion of *hypothesis set stability* that we introduce in this paper. This notion admits algorithmic stability as a special case, when the hypotheses sets are reduced to singletons. We also introduce a new notion of *Rademacher complexity for data-dependent hypothesis sets*. Our main results are generalization bounds (Section 4) for stable data-dependent hypothesis sets expressed in terms of the hypothesis set stability parameter, our notion of Rademacher complexity, and a notion of *cross-validation stability* that, in turn, can be upper-bounded by the diameter of the family of hypothesis sets. Our learning bounds admit as special cases both standard Rademacher complexity bounds and algorithm-dependent uniform stability bounds.

**4. New generalization bounds for specific learning applications.** In section 5 (see also Appendix G), we illustrate the generality and the benefits of our hypothesis set stability learning bounds by using them to derive new generalization bounds for several learning applications. To our knowledge, there is no straightforward analysis based on previously existing tools that yield these generalization bounds. These applications include: (a) *bagging* algorithms that may employ non-uniform, data-dependent, averaging of the base predictors, (b) *stochastic strongly-convex optimization* algorithms based on an average of other stochastic optimization algorithms, (c) *stable representation* learning algorithms, which first learn a data representation using the sample and then learn a predictor on top of the learned representation, and (d) *distillation* algorithms, which first compute a complex predictor using the sample and then use it to learn a simpler predictor that is close to it.

## 1.2 Related work on data-dependent hypothesis sets.

Shawe-Taylor et al. [1998] presented an analysis of structural risk minimization over data-dependent hierarchies based on a concept of *luckiness*, which generalizes the notion of margin of linear classifiers. Their analysis can be viewed as an alternative and general study of data-dependent hypothesis sets,

using luckiness functions and $\omega$-smallness (or $\omega$-smoothness) conditions. A luckiness function helps decompose a hypothesis set into *lucky sets*, that is sets of functions *luckier* than a given function. The luckiness framework is attractive and the notion of luckiness, for example margin, can in fact be combined with our results. However, finding pairs of truly data-dependent luckiness and $\omega$-smallness functions, other than those based on the margin and the empirical VC-dimension, is quite difficult, in particular because of the very technical $\omega$-smallness condition [see Philips, 2005, p. 70]. In contrast, hypothesis set stability is simpler and often easier to bound. The notions of luckiness and $\omega$-smallness have also been used by Herbrich and Williamson [2002] to derive algorithm-specific guarantees. In fact, the authors show a connection with algorithmic stability (not hypothesis set stability), at the price of a guarantee requiring the strong condition that the stability parameter be in $o(1/m)$, where $m$ is the sample size [see Herbrich and Williamson, 2002, pp. 189-190].

Data-dependent hypothesis classes are conceptually related to the notion of data-dependent priors in PAC-Bayesian generalization bounds. Catoni [2007] developed localized PAC-Bayes analysis by using a prior defined in terms of the data generating distribution. This work was extended by Lever et al. [2013] who proved sharp risk bounds for stochastic exponential weights algorithms. Parrado-Hernández et al. [2012] investigated the possibility of learning the prior from a separate data set, as well as priors obtained via computing a data-dependent bound on the KL term. More closely related to this paper is the work of Dziugaite and Roy [2018a,b], who develop PAC-Bayes bounds by choosing the prior via a data-dependent differentially private mechanism, and also showed that weaker notions than differential privacy also suffice to yield valid bounds. In Appendix H, we give a more detailed discussion of PAC-Bayes bounds, in particular to show how finer PAC-Bayes bounds than standard ones can be derived from Rademacher complexity bounds, here with an alternative analysis and constants than [Kakade et al., 2008] and how data-dependent PAC-Bayes bounds can be derived from our data-dependent Rademacher complexity bounds. More discussion on data-dependent priors can be found in Appendix F.3.

## 2 Definitions and Properties

Let $\mathcal{X}$ be the input space and $\mathcal{Y}$ the output space, and define $\mathcal{Z} \coloneqq \mathcal{X} \times \mathcal{Y}$ We denote by $\mathcal{D}$ the unknown distribution over $\mathcal{X} \times \mathcal{Y}$ according to which samples are drawn.

The hypotheses $h$ we consider map $\mathcal{X}$ to a set $\mathcal{Y}'$ sometimes different from $\mathcal{Y}$. For example, in binary classification, we may have $\mathcal{Y} = \{-1, +1\}$ and $\mathcal{Y}' = \mathbb{R}$. Thus, we denote by $\ell \colon \mathcal{Y}' \times \mathcal{Y} \to [0, 1]$ a loss function defined on $\mathcal{Y}' \times \mathcal{Y}$ and taking non-negative real values bounded by one. We denote the loss of a hypothesis $h \colon \mathcal{X} \to \mathcal{Y}'$ at point $z = (x, y) \in \mathcal{X} \times \mathcal{Y}$ by $L(h, z) = \ell(h(x), y)$. We denote by $R(h)$ the generalization error or expected loss of a hypothesis $h \in \mathcal{H}$ and by $\widehat{R}_S(h)$ its empirical loss over a sample $S = (z_1, \ldots, z_m)$:

$$R(h) = \mathop{\mathbb{E}}_{z \sim \mathcal{D}}[L(h, z)] \qquad \widehat{R}_S(h) = \mathop{\mathbb{E}}_{z \sim S}[L(h, z)] = \frac{1}{m} \sum_{i=1}^{m} L(h, z_i).$$

In the general framework we consider, a hypothesis set depends on the sample received. We will denote by $\mathcal{H}_S$ the hypothesis set depending on the labeled sample $S \in \mathcal{Z}^m$ of size $m \geq 1$. We assume that $\mathcal{H}_S$ is invariant to the ordering of the points in $S$.

**Definition 1** (Hypothesis set uniform stability). *Fix $m \geq 1$. We will say that a family of data-dependent hypothesis sets $\mathcal{H} = (\mathcal{H}_S)_{S \in \mathcal{Z}^m}$ is $\beta$-uniformly stable (or simply $\beta$-stable) for some $\beta \geq 0$, if for any two samples $S$ and $S'$ of size $m$ differing only by one point, the following holds:*

$$\forall h \in \mathcal{H}_S, \exists h' \in \mathcal{H}_{S'} \colon \forall z \in \mathcal{Z}, |L(h, z) - L(h', z)| \leq \beta. \tag{1}$$

Thus, two hypothesis sets derived from samples differing by one element are close in the sense that any hypothesis in one admits a counterpart in the other set with $\beta$-similar losses. A closely related notion is the *sensitivity* of a function $f \colon \mathcal{Z}^m \to \mathbb{R}$. Such a function $f$ is called $\beta$-sensitive if for any two samples $S$ and $S'$ of size $m$ differing only by one point, we have $|f(S) - f(S')| \leq \beta$.

We also introduce a new notion of Rademacher complexity for data-dependent hypothesis sets. To introduce its definition, for any two samples $S, T \in \mathcal{Z}^m$ and a vector of Rademacher variables $\boldsymbol{\sigma}$, denote by $S_{T, \boldsymbol{\sigma}}$ the sample derived from $S$ by replacing its $i$th element with the $i$th element of $T$, for all $i \in [m] = \{1, 2, \ldots, m\}$ with $\sigma_i = -1$. We will use $\mathcal{H}_{S,T}^{\boldsymbol{\sigma}}$ to denote the hypothesis set $\mathcal{H}_{S_{T,\boldsymbol{\sigma}}}$.

**Definition 2** (Rademacher complexity of data-dependent hypothesis sets). *Fix $m \geq 1$. The empirical Rademacher complexity $\widehat{\mathfrak{R}}_{S,T}^\diamond(\mathcal{H})$ and the Rademacher complexity $\mathfrak{R}_m^\diamond(\mathcal{H})$ of a family of data-dependent hypothesis sets $\mathcal{H} = (\mathcal{H}_S)_{S \in \mathcal{Z}^m}$ for two samples $S = (z_1^S, \ldots, z_m^S)$ and $T = (z_1^T, \ldots, z_m^T)$ in $\mathcal{Z}^m$ are defined by*

$$\widehat{\mathfrak{R}}_{S,T}^\diamond(\mathcal{H}) = \frac{1}{m} \mathbb{E}_{\boldsymbol{\sigma}}\left[\sup_{h \in \mathcal{H}_{S,T}^{\boldsymbol{\sigma}}} \sum_{i=1}^m \sigma_i h(z_i^T)\right] \qquad \mathfrak{R}_m^\diamond(\mathcal{H}) = \frac{1}{m} \mathbb{E}_{\substack{S,T \sim \mathcal{D}^m \\ \boldsymbol{\sigma}}}\left[\sup_{h \in \mathcal{H}_{S,T}^{\boldsymbol{\sigma}}} \sum_{i=1}^m \sigma_i h(z_i^T)\right]. \qquad (2)$$

When the family of data-dependent hypothesis sets $\mathcal{H}$ is $\beta$-stable with $\beta = O(1/m)$, the empirical Rademacher complexity $\widehat{\mathfrak{R}}_{S,T}^\diamond(\mathcal{G})$ is sharply concentrated around its expectation $\mathfrak{R}_m^\diamond(\mathcal{G})$, as with the standard empirical Rademacher complexity (see Lemma 4).

Let $\mathcal{H}_{S,T}$ denote the union of all hypothesis sets based on $\mathcal{U} = \{U : U \subseteq (S \cup T), U \in \mathcal{Z}^m\}$, the subsamples of $S \cup T$ of size $m$: $\mathcal{H}_{S,T} = \bigcup_{U \in \mathcal{U}} \mathcal{H}_U$. Since for any $\boldsymbol{\sigma}$, we have $\mathcal{H}_{S,T}^{\boldsymbol{\sigma}} \subseteq \mathcal{H}_{S,T}$, the following simpler upper bound in terms of the standard empirical Rademacher complexity of $\mathcal{H}_{S,T}$ can be used for our notion of empirical Rademacher complexity:

$$\mathfrak{R}_m^\diamond(\mathcal{H}) \leq \frac{1}{m} \mathbb{E}_{\substack{S,T \sim \mathcal{D}^m \\ \boldsymbol{\sigma}}}\left[\sup_{h \in \mathcal{H}_{S,T}} \sum_{i=1}^m \sigma_i h(z_i^T)\right] = \mathbb{E}_{S,T \sim \mathcal{D}^m}\left[\widehat{\mathfrak{R}}_T(\mathcal{H}_{S,T})\right],$$

where $\widehat{\mathfrak{R}}_T(\mathcal{H}_{S,T})$ is the standard empirical Rademacher[1] complexity of $\mathcal{H}_{S,T}$ for the sample $T$. Some properties of our notion of Rademacher complexity are given in Appendix B.

Let $\mathcal{G}_S$ denote the family of loss functions associated to $\mathcal{H}_S$:

$$\mathcal{G}_S = \{z \mapsto L(h,z) : h \in \mathcal{H}_S\}, \qquad (3)$$

and let $\mathcal{G} = (\mathcal{G}_S)_{S \in \mathcal{Z}^m}$ denote the family of hypothesis sets $\mathcal{G}_S$. Our main results will be expressed in terms of $\mathfrak{R}_m^\diamond(\mathcal{G})$. When the loss function $\ell$ is $\mu$-Lipschitz, by Talagrand's contraction lemma [Ledoux and Talagrand, 1991], in all our results, $\mathfrak{R}_m^\diamond(\mathcal{G})$ can be replaced by $\mu \mathbb{E}_{S,T \sim \mathcal{D}^m}[\widehat{\mathfrak{R}}_T(\mathcal{H}_{S,T})]$.

Rademacher complexity is one way to measure the capacity of the family of data-dependent hypothesis sets. We also derive learning bounds in situations where a notion of *diameter* of the hypothesis sets is small. We now define a notion of *cross-validation stability* and diameter for data-dependent hypothesis sets. In the following, for a sample $S$, $S^{z \leftrightarrow z'}$ denotes the sample obtained from $S$ by replacing $z \in S$ by $z' \in \mathcal{Z}$.

**Definition 3** (Hypothesis set Cross-Validation (CV) stability, diameter). *Fix $m \geq 1$. For some $\bar{\chi}, \chi, \bar{\Delta}, \Delta, \Delta_{\max} \geq 0$, we say that a family of data-dependent hypothesis sets $\mathcal{H} = (\mathcal{H}_S)_{S \in \mathcal{Z}^m}$ has average CV-stability $\bar{\chi}$, CV-stability $\chi$, average diameter $\bar{\Delta}$, diameter $\Delta$ and max-diameter $\Delta_{\max}$ if the following hold:*

$$\mathbb{E}_{S \sim \mathcal{D}^m} \mathbb{E}_{z' \sim \mathcal{D}, z \sim S}\left[\sup_{h \in \mathcal{H}_S, h' \in \mathcal{H}_{S^{z \leftrightarrow z'}}} L(h',z) - L(h,z)\right] \leq \bar{\chi} \qquad (4)$$

$$\sup_{S \in \mathcal{Z}^m} \mathbb{E}_{z' \sim \mathcal{D}, z \sim S}\left[\sup_{h \in \mathcal{H}_S, h' \in \mathcal{H}_{S^{z \leftrightarrow z'}}} L(h',z) - L(h,z)\right] \leq \chi \qquad (5)$$

$$\mathbb{E}_{S \sim \mathcal{D}^m} \mathbb{E}_{z \sim S}\left[\sup_{h, h' \in \mathcal{H}_S} L(h',z) - L(h,z)\right] \leq \bar{\Delta} \qquad (6)$$

$$\sup_{S \in \mathcal{Z}^m} \mathbb{E}_{z \sim S}\left[\sup_{h, h' \in \mathcal{H}_S} L(h',z) - L(h,z)\right] \leq \Delta \qquad (7)$$

$$\sup_{S \in \mathcal{Z}^m} \max_{z \in S}\left[\sup_{h, h' \in \mathcal{H}_S} L(h',z) - L(h,z)\right] \leq \Delta_{\max}. \qquad (8)$$

CV-stability of hypothesis sets can be bounded in terms of their stability and diameter (see straightforward proof in Appendix C).

**Lemma 1.** *Suppose a family of data-dependent hypothesis sets $\mathcal{H}$ is $\beta$-uniformly stable. Then if it has diameter $\Delta$, then it is $(\Delta + \beta)$-CV-stable, and if it has average diameter $\bar{\Delta}$ then it is $(\bar{\Delta} + \beta)$-average CV-stable.*

## 3 General learning bound for data-dependent hypothesis sets

In this section, we present general learning bounds for data-dependent hypothesis sets that do not make use of the notion of hypothesis set stability. One straightforward idea to derive such guarantees for data-dependent hypothesis sets is to replace the hypothesis set $\mathcal{H}_S$ depending on the observed sample $S$ by the union of all such hypothesis sets over all samples of size $m$, $\overline{\mathcal{H}}_m = \bigcup_{S \in \mathcal{Z}^m} \mathcal{H}_S$. However, in general, $\overline{\mathcal{H}}_m$ can be very rich, which can lead to uninformative learning bounds. A somewhat better alternative consists of considering the union of all such hypothesis sets for samples of size $m$ included in some supersample $U$ of size $m + n$, with $n \geq 1$, $\overline{\mathcal{H}}_{U,m} = \bigcup_{\substack{S \in \mathcal{Z}^m \\ S \subseteq U}} \mathcal{H}_S$. We will

derive learning guarantees based on the maximum *transductive Rademacher complexity* of $\overline{\mathcal{H}}_{U,m}$. There is a trade-off in the choice of $n$: smaller values lead to less complex sets $\overline{\mathcal{H}}_{U,m}$, but they also lead to weaker dependencies on sample sizes. Our bounds are more refined guarantees than the shattering-coefficient bounds originally given for this problem by Gat [2001] in the case $n = m$, and later by Cannon et al. [2002] for any $n \geq 1$. They also apply to arbitrary bounded loss functions and not just the binary loss. They are expressed in terms of the following notion of *transductive Rademacher complexity for data-dependent hypothesis sets*:

$$\widehat{\mathfrak{R}}^{\diamond}_{U,m}(\mathcal{G}) = \mathbb{E}_{\boldsymbol{\sigma}} \left[ \sup_{h \in \overline{\mathcal{H}}_{U,m}} \frac{1}{m+n} \sum_{i=1}^{m+n} \sigma_i L(h, z_i^U) \right],$$

where $U = (z_1^U, \dots, z_{m+n}^U) \in \mathcal{Z}^{m+n}$ and where $\boldsymbol{\sigma}$ is a vector of $(m+n)$ independent random variables taking value $\frac{m+n}{n}$ with probability $\frac{n}{m+n}$, and $-\frac{m+n}{m}$ with probability $\frac{m}{m+n}$. Our notion of transductive Rademacher complexity is simpler than that of El-Yaniv and Pechyony [2007] (in the data-independent case) and leads to simpler proofs and guarantees. A by-product of our analysis is learning guarantees for standard transductive learning in terms of this notion of transductive Rademacher complexity, which can be of independent interest.

**Theorem 1.** *Let $\mathcal{H} = (\mathcal{H}_S)_{S \in \mathcal{Z}^m}$ be a family of data-dependent hypothesis sets. Let $\mathcal{G}$ be defined as in (3). Then, for any $\delta > 0$, with probability at least $1 - \delta$ over the choice of the draw of the sample $S \sim \mathcal{Z}^m$, the following inequality holds for all $h \in \mathcal{H}_S$:*

$$R(h) \leq \widehat{R}_S(h) + \max_{U \in \mathcal{Z}^{m+n}} 2\widehat{\mathfrak{R}}^{\diamond}_{U,m}(\mathcal{G}) + 3\sqrt{\left(\frac{1}{m} + \frac{1}{n}\right)\log(\frac{2}{\delta})} + 2\sqrt{\left(\frac{1}{m} + \frac{1}{n}\right)^3 mn}.$$

*Proof.* (Sketch; full proof in Appendix D.) We use the following symmetrization result, which holds for any $\epsilon > 0$ with $m\epsilon^2 \geq 2$ for data-dependent hypothesis sets (Lemma 5, Appendix D):

$$\mathbb{P}_{S \sim \mathcal{D}^m}\left[ \sup_{h \in \mathcal{H}_S} R(h) - \widehat{R}_S(h) > \epsilon \right] \leq 2 \mathbb{P}_{\substack{S \sim \mathcal{D}^m \\ T \sim \mathcal{D}^n}}\left[ \sup_{h \in \mathcal{H}_S} \widehat{R}_T(h) - \widehat{R}_S(h) > \frac{\epsilon}{2} \right].$$

To bound the right-hand side, we use an extension of McDiarmid's inequality to sampling without replacement [Cortes et al., 2008] applied to $\Phi(S) = \sup_{h \in \overline{\mathcal{H}}_{U,m}} \widehat{R}_T(h) - \widehat{R}_S(h)$. Lemma 6 (Appendix D) is then used to bound $\mathbb{E}[\Phi(S)]$ in terms of our notion of transductive Rademacher complexity. $\quad\square$

## 4 Learning bound for stable data-dependent hypothesis sets

In this section, we present our main generalization bounds for data-dependent hypothesis sets.

**Theorem 2.** *Let $\mathcal{H} = (\mathcal{H}_S)_{S \in \mathcal{Z}^m}$ be a $\beta$-stable family of data-dependent hypothesis sets, with $\bar{\chi}$ average CV-stability, $\chi$ CV-stability and $\Delta_{\max}$ max-diameter. Let $\mathcal{G}$ be defined as in (3). Then, for any $\delta > 0$, with probability at least $1 - \delta$ over the draw of a sample $S \sim \mathcal{Z}^m$, the following inequality*

*holds for all $h \in \mathcal{H}_S$:*

$$\forall h \in \mathcal{H}_S, R(h) \leq \widehat{R}_S(h) + \min \Bigg\{ \min \left\{ 2\mathfrak{R}_m^\diamond(\mathcal{G}), \bar{\chi} \right\} + (1 + 2\beta m)\sqrt{\tfrac{1}{2m}\log(\tfrac{1}{\delta})}, \tag{9}$$

$$\sqrt{e}\chi + 4\sqrt{(\tfrac{1}{m} + 2\beta)\log(\tfrac{6}{\delta})}, \tag{10}$$

$$48(3\beta + \Delta_{\max})\log(m)\log(\tfrac{5m^3}{\delta}) + \sqrt{\tfrac{4}{m}\log(\tfrac{4}{\delta})} \Bigg\}. \tag{11}$$

The proof of the theorem is given in Appendix E. The main idea is to control the sensitivity of the function $\Psi(S, S')$ defined for any two samples $S, S'$, as follows:

$$\Psi(S, S') = \sup_{h \in \mathcal{H}_S} R(h) - \widehat{R}_{S'}(h).$$

To prove bound (9), we apply McDiarmid's inequality to $\Psi(S, S)$, using the $(\tfrac{1}{m} + 2\beta)$-sensitivity of $\Psi(S, S)$, and then upper bound the expectation $\mathbb{E}_{S \sim \mathcal{D}^m}[\Psi(S, S)]$ in terms of our notion of Rademacher complexity. The bound (10) is obtained via a differential-privacy-based technique, as in Feldman and Vondrak [2018], and (11) is a direct consequence of the bound of Feldman and Vondrak [2019] using the observation that an algorithm that chooses an arbitrary $h \in \mathcal{H}_S$ is $O(\beta + \Delta_{\max})$-uniformly stable in the classical [Bousquet and Elisseeff, 2002] sense.

Bound (9) admits as a special case the standard Rademacher complexity bound for fixed hypothesis sets [Koltchinskii and Panchenko, 2002, Bartlett and Mendelson, 2002]: in that case, we have $\mathcal{H}_S = \mathcal{H}$ for some $\mathcal{H}$, thus $\mathfrak{R}_m^\diamond(\mathcal{G})$ coincides with the standard Rademacher complexity $\mathfrak{R}_m(\mathcal{G})$; furthermore, the family of hypothesis sets is 0-stable, thus the bound holds with $\beta = 0$.

Bounds (10) and (11) specialize to the bounds of Feldman and Vondrak [2018] and Feldman and Vondrak [2019] respectively for the special case of standard uniform stability of algorithms, since in that case, $\mathcal{H}_S$ is reduced to a singleton, $\mathcal{H}_S = \{h_S\}$, and so $\Delta = 0$, which implies that $\chi \leq \Delta + \beta = \beta$.

The Rademacher complexity-based bound (9) typically gives the tightest control on generalization error compared to the bounds (10) and (11), which rely on the cruder diameter notion. However the diameter may be easier to bound for some applications than the Rademacher complexity. To compare the diameter-based bounds, in applications where $\Delta_{\max} = O(\Delta)$, bound (11) may be tighter than (10). But, in several applications, we have $\beta = O(\tfrac{1}{m})$, and then bound (10) is tighter.

## 5 Applications

We now discuss several applications of our learning guarantees, with some others in Appendix G.

### 5.1 Bagging

*Bagging* [Breiman, 1996] is a prominent ensemble method used to improve the stability of learning algorithms. It consists of generating $k$ new samples $B_1, B_2, \ldots, B_k$, each of size $p$, by sampling uniformly with replacement from the original sample $S$ of size $m$. An algorithm $\mathcal{A}$ is then trained on each of these samples to generate $k$ predictors $\mathcal{A}(B_i)$, $i \in [k]$. In regression, the predictors are combined by taking a convex combination $\sum_{i=1}^k w_i \mathcal{A}(B_i)$. Here, we analyze a common instance of bagging to illustrate the application of our learning guarantees: we will assume a regression setting and a uniform sampling from $S$ *without replacement*.[2] We will also assume that the loss function is $\mu$-Lipschitz in its first argument, that the predictions are in the range $[0, 1]$, and that all the mixing weights $w_i$ are bounded by $\tfrac{C}{k}$ for some constant $C \geq 1$, in order to ensure that no subsample $B_i$ is overly influential in the final regressor (in practice, a uniform mixture is typically used in bagging).

To analyze bagging, we cast it in our framework. First, to deal with the randomness in choosing the subsamples, we can equivalently imagine the process as choosing *indices* in $[m]$ to form the subsamples rather than samples in $S$, and then once $S$ is drawn, the subsamples are generated by

filling in the samples at the corresponding indexes. For any index $i \in [m]$, the chance that it is picked in any subsample is $\frac{p}{m}$. Thus, by Chernoff's bound, with probability at least $1 - \delta$, no index in $[m]$ appears in more than $t := \frac{kp}{m} + \sqrt{\frac{2kp \log(\frac{m}{\delta})}{m}}$ subsamples. In the following, we condition on the random seed of the bagging algorithm so that this is indeed the case, and later use a union bound to control the chance that the chosen random seed does not satisfy this property, as elucidated in section F.2.

Define the data-dependent family of hypothesis sets $\mathcal{H}$ as $\mathcal{H}_S := \left\{ \sum_{i=1}^{k} w_i \mathcal{A}(B_i) \colon w \in \Delta_k^{C/k} \right\}$, where $\Delta_k^{C/k}$ denotes the simplex of distributions over $k$ items with all weights $w_i \leq \frac{C}{k}$. Next, we give upper bounds on the hypothesis set stability and the Rademacher complexity of $\mathcal{H}$. Assume that algorithm $\mathcal{A}$ admits uniform stability $\beta_A$ [Bousquet and Elisseeff, 2002], i.e. for any two samples $B$ and $B'$ of size $p$ that differ in exactly one data point and for all $x \in \mathcal{X}$, we have $|\mathcal{A}(B)(x) - \mathcal{A}(B')(x)| \leq \beta_A$. Now, let $S$ and $S'$ be two samples of size $m$ differing by one point at the same index, $z \in S$ and $z' \in S'$. Then, consider the subsets $B_i'$ of $S'$ which are obtained from the $B_i$s by copying over all the elements except $z$, and replacing all instances of $z$ by $z'$. For any $B_i$, if $z \notin B_i$, then $\mathcal{A}(B_i) = \mathcal{A}(B_i')$ and, if $z \in B_i$, then $|\mathcal{A}(B_i)(x) - \mathcal{A}(B_i')(x)| \leq \beta_A$ for any $x \in \mathcal{X}$. We can now bound the hypothesis set uniform stability as follows: since $L$ is $\mu$-Lipschitz in the prediction, for any $z'' \in \mathcal{Z}$, and any $w \in \Delta_k^{C/k}$, we have

$$\left| L\left(\sum_{i=1}^{k} w_i \mathcal{A}(B_i), z''\right) - L\left(\sum_{i=1}^{k} w_i \mathcal{A}(B_i'), z''\right) \right| \leq \left[ \frac{p}{m} + \sqrt{\frac{2p \log(\frac{1}{\delta})}{km}} \right] \cdot C \mu \beta_A.$$

It is easy to check the CV-stability and diameter of the hypothesis sets is $\Omega(1)$ in the worst case. Thus, the CV-stability-based bound (10) and standard uniform-stability bound (11) are not informative here, and we use the Rademacher complexity based bound (9) instead. Bounding the Rademacher complexity $\widehat{\mathfrak{R}}_S(\mathcal{H}_{S,T})$ for $S, T \in \mathcal{Z}^m$ is non-trivial. Instead, we can derive a reasonable upper bound by analyzing the Rademacher complexity of a larger function class. Specifically, for any $z \in \mathcal{Z}$, define the $d := \binom{2m}{p}$-dimensional vector $u_z = \langle \mathcal{A}(B)(z) \rangle_{B \subseteq S \cup T, |B|=p}$. Then, the class of functions is $\mathcal{F}_{S,T} := \{ z \mapsto w^\top u_z \colon w \in \mathbb{R}^d, \|w\|_1 = 1 \}$. Clearly $\mathcal{H}_{S,T} \subseteq \mathcal{F}_{S,T}$. Since $\|u_z\|_\infty \leq 1$, a standard Rademacher complexity bound (see Theorem 11.15 in [Mohri et al., 2018]) implies $\widehat{\mathfrak{R}}_S(\mathcal{F}_{S,T}) \leq \sqrt{\frac{2 \log\left(2\binom{2m}{p}\right)}{m}} \leq \sqrt{\frac{2p \log(4m)}{m}}$. Thus, by Talagrand's inequality, we conclude that $\widehat{\mathfrak{R}}_S(\mathcal{G}_{S,T}) \leq \mu \sqrt{\frac{2p \log(4m)}{m}}$. In view of that, by Theorem 2, for any $\delta > 0$, with probability at least $1 - 2\delta$ over the draws of a sample $S \sim \mathcal{D}^m$ and the randomness in the bagging algorithm, the following inequality holds for any $h \in \mathcal{H}_S$:

$$R(h) \leq \widehat{R}_S(h) + 2\mu \sqrt{\frac{2p \log(4m)}{m}} + \left[ 1 + 2\left[ p + \sqrt{\frac{2pm \log(\frac{1}{\delta})}{k}} \right] \cdot C \mu \beta_A \right] \sqrt{\frac{\log \frac{2}{\delta}}{2m}}.$$

For $p = o(\sqrt{m})$ and $k = \omega(p)$, the generalization gap goes to 0 as $m \to \infty$, *regardless* of the stability of $\mathcal{A}$. This gives a new generalization guarantee for bagging, similar (but incomparable) to the one derived by Elisseeff et al. [2005]. One major point of difference is that unlike their bound, our bound allows for non-uniform averaging schemes.

## 5.2 Stochastic strongly-convex optimization

Here, we consider data-dependent hypothesis sets based on stochastic strongly-convex optimization algorithms. As shown by Shalev-Shwartz et al. [2010], uniform convergence bounds do not hold for the stochastic convex optimization problem in general.

Consider $K$ stochastic strongly-convex optimization algorithms $\mathcal{A}_j$, each returning vector $\widehat{w}_j^S$, after receiving sample $S \in \mathcal{Z}^m$, $j \in [K]$. As shown by Shalev-Shwartz et al. [2010], such algorithms are $\beta = O(\frac{1}{m})$ sensitive in their output vector, i.e. for all $j \in [K]$, we have $\|\widehat{w}_j^S - \widehat{w}_j^{S'}\| \leq \beta$ if $S$ and $S'$ differ by one point.

Assume that the loss $L(w, z)$ is $\mu$-Lipschitz with respect to its first argument $w$. Let the data-dependent hypothesis set be defined as follows: $\mathcal{H}_S = \left\{ \sum_{j=1}^{K} \alpha_j \widehat{w}_j^S \colon \alpha \in \Delta_K \cap \mathsf{B}_1(\alpha_0, r) \right\}$, where

$\alpha_0$ is in the simplex of distributions $\Delta_K$ and $\mathsf{B}_1(\alpha_0, r)$ is the $L_1$ ball of radius $r > 0$ around $\alpha_0$. We choose $r = \frac{1}{2\mu D \sqrt{m}}$. A natural choice for $\alpha_0$ would be the uniform mixture.

Since the loss function is $\mu$-Lipschitz, the family of hypotheses $\mathcal{H}_S$ is $\mu\beta$-stable. In this setting, bounding the Rademacher complexity is difficult, so we resort to the diameter based bound (10) instead. Note that for any $\alpha, \alpha' \in \Delta_K \cap \mathsf{B}_1(\alpha_0, r)$ and any $z \in \mathcal{Z}$, we have

$$L\left(\sum_{j=1}^{K} \alpha_j \widehat{w}_j^S, z\right) - L\left(\sum_{j=1}^{K} \alpha_j' \widehat{w}_j^S, z\right) \leq \mu \left\|\sum_{j=1}^{K}(\alpha_i - \alpha_j')\widehat{w}_j^S\right\|_2 \leq \mu \left\|[w_1^S \cdots w_K^S]\right\|_{1,2} \|\alpha - \alpha'\|_1 \leq 2\mu r D,$$

where $\left\|[w_1^S \cdots w_K^S]\right\|_{1,2} := \max_{x \neq 0} \frac{\|\sum_{j=1}^{k} x_j w_j^S\|_2}{\|x\|_1} = \max_{i \in [K]} \|w_i^S\|_2 \leq D$. Thus, the average diameter admits the following upper bound: $\widehat{\Delta} \leq 2\mu r D = \frac{1}{\sqrt{m}}$. In view of that, by Theorem 2, for any $\delta > 0$, with probability at least $1 - \delta$, the following holds for all $\alpha \in \Delta_K \cap \mathsf{B}_1(\alpha_0, r)$:

$$\mathbb{E}_{z \sim \mathcal{D}}\left[L\left(\sum_{j=1}^{K} \alpha_j \widehat{w}_j^S, z\right)\right] \leq \frac{1}{m}\sum_{i=1}^{m} L\left(\sum_{j=1}^{K} \alpha_i \widehat{w}_j^S, z_i^S\right) + \sqrt{\frac{e}{m}} + \sqrt{e}\mu\beta + 4\sqrt{\left(\frac{1}{m} + 2\mu\beta\right)\log\left(\frac{6}{\delta}\right)}.$$

The second stage of an algorithm in this context consists of choosing $\alpha$, potentially using a non-stable algorithm. This application both illustrates the use of our learning bounds using the diameter and its application even in the absence of uniform convergence bounds.

As an aside, we note that the analysis of section 5.1 can be carried over to this setting, by setting $\mathcal{A}$ to be a stochastic strongly-convex optimization algorithm which outputs a weight vector $\hat{w}$. This yields generalization bounds for aggregating over a larger set of mixing weights, albeit with the restriction that each algorithm uses only a small part of $S$.

### 5.3 $\Delta$-sensitive feature mappings

Consider the scenario where the training sample $S \in \mathcal{Z}^m$ is used to learn a non-linear feature mapping $\Phi_S \colon \mathcal{X} \to \mathbb{R}^N$ that is $\Delta$-sensitive for some $\Delta = O(\frac{1}{m})$. $\Phi_S$ may be the feature mapping corresponding to some positive definite symmetric kernel or a mapping defined by the top layer of an artificial neural network trained on $S$, with a stability property.

To define the second state, let $\mathcal{L}$ be a set of $\gamma$-Lipschitz functions $f \colon \mathbb{R}^N \to \mathbb{R}$. Then we define $\mathcal{H}_S = \{x \mapsto f(\Phi_S(x)) \colon f \in \mathcal{L}\}$. Assume that the loss function $\ell$ is $\mu$-Lipschitz with respect to its first argument. Then, for any hypothesis $h \colon x \mapsto f(\Phi_S(x)) \in \mathcal{H}_S$ and any sample $S'$ differing from $S$ by one element, the hypothesis $h' \colon x \mapsto f(\Phi_{S'}(x)) \in \mathcal{H}_{S'}$ admits losses that are $\beta$-close to those of $h$, with $\beta = \mu\gamma\Delta$, since, for all $(x, y) \in \mathcal{X} \times \mathcal{Y}$, by the Cauchy-Schwarz inequality, the following inequality holds:

$$\ell(f(\Phi_S(x)), y) - \ell(f(\Phi_{S'}(x)), y) \leq \mu|f((\Phi_S(x)) - f(\Phi_{S'}(x))| \leq \mu\gamma\|\Phi_S(x) - \Phi_{S'}(x)\| \leq \mu\gamma\Delta.$$

Thus, the family of hypothesis set $\mathcal{H} = (\mathcal{H}_S)_{S \in \mathcal{Z}^m}$ is uniformly $\beta$-stable with $\beta = \mu\gamma\Delta = O(\frac{1}{m})$. In view of that, by Theorem 2, for any $\delta > 0$, with probability at least $1 - \delta$ over the draw of a sample $S \sim \mathcal{D}^m$, the following inequality holds for any $h \in \mathcal{H}_S$:

$$R(h) \leq \widehat{R}_S(h) + 2\mathfrak{R}_m^\diamond(\mathcal{G}) + (1 + 2\mu\gamma\Delta m)\sqrt{\frac{1}{2m}\log(\frac{1}{\delta})}. \tag{12}$$

Notice that this bound applies even when the second stage of an algorithm, which consists of selecting a hypothesis $h_S$ in $\mathcal{H}_S$, is not stable. A standard uniform stability guarantee cannot be used in that case. The setting described here can be straightforwardly extended to the case of other norms for the definition of sensitivity and that of the norm used in the definition of $\mathcal{H}_S$.

### 5.4 Distillation

Here, we consider *distillation algorithms* which, in the first stage, train a very complex model on the labeled sample. Let $f_S^* \colon \mathcal{X} \to \mathbb{R}$ denote the resulting predictor for a training sample $S$ of size $m$. We will assume that the training algorithm is $\beta$-sensitive, that is $\|f_S^* - f_{S'}^*\| \leq \beta = O(\frac{1}{m})$ for $S$ and $S'$ differing by one point.

In the second stage, the algorithm selects a hypothesis that is $\gamma$-close to $f_S^*$ from a less complex family of predictors $\mathcal{H}$. This defines the following sample-dependent hypothesis set: $\mathcal{H}_S = \{h \in \mathcal{H} : \|(h - f_S^*)\|_\infty \leq \gamma\}$.

Assume that the loss $\ell$ is $\mu$-Lipschitz with respect to its first argument and that $\mathcal{H}$ is a subset of a vector space. Let $S$ and $S'$ be two samples differing by one point. Note, $f_S^*$ may not be in $\mathcal{H}$, but we will assume that $f_{S'}^* - f_S^*$ is in $\mathcal{H}$. Let $h$ be in $\mathcal{H}_S$, then the hypothesis $h' = h + f_{S'}^* - f_S^*$ is in $\mathcal{H}_{S'}$ since $\|h' - f_{S'}^*\|_\infty = \|h - f_S^*\|_\infty \leq \gamma$. Figure 2 illustrates the hypothesis sets. By the $\mu$-Lipschitzness of the loss, for any $z = (x, y) \in \mathcal{Z}$, $|\ell(h'(x), y) - \ell(h(x), y)| \leq \mu\|h'(x) - h(x)\|_\infty = \mu\|f_{S'}^* - f_S^*\| \leq \mu\beta$. Thus, the family of hypothesis sets $\mathcal{H}_S$ is $\mu\beta$-stable.

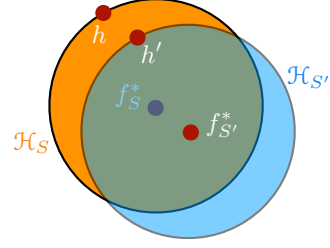

Figure 2: Illustration of the distillation hypothesis sets. Notice that the diameter of a hypothesis set $\mathcal{H}_S$ may be large here.

In view of that, by Theorem 2, for any $\delta > 0$, with probability at least $1 - \delta$ over the draw of a sample $S \sim \mathcal{D}^m$, the following inequality holds for any $h \in \mathcal{H}_S$:

$$R(h) \leq \widehat{R}_S(h) + 2\mathfrak{R}_m^\diamond(\mathcal{G}) + (1 + 2\mu\beta m)\sqrt{\tfrac{1}{2m}\log(\tfrac{1}{\delta})}.$$

Notice that a standard uniform-stability argument would not necessarily apply here since $\mathcal{H}_S$ could be relatively complex and the second stage not necessarily stable.

## 6   Conclusion

We presented a broad study of generalization with data-dependent hypothesis sets, including general learning bounds using a notion of transductive Rademacher complexity and, more importantly, learning bounds for stable data-dependent hypothesis sets. We illustrated the applications of these guarantees to the analysis of several problems. Our framework is general and covers learning scenarios commonly arising in applications for which standard generalization bounds are not applicable. Our results can be further augmented and refined to include model selection bounds and local Rademacher complexity bounds for stable data-dependent hypothesis sets (to be presented in a more extended version of this manuscript), and further extensions described in Appendix F. Our analysis can also be extended to the non-i.i.d. setting and other learning scenarios such as that of transduction. Several by-products of our analysis, including our proof techniques, new guarantees for transductive learning, and our PAC-Bayesian bounds for randomized algorithms, both in the sample-independent and sample-dependent cases, can be of independent interest. While we highlighted several applications of our learning bounds, a tighter analysis might be needed to derive guarantees for a wider range of data-dependent hypothesis classes or scenarios.

**Acknowledgements.**   HL is supported by NSF IIS-1755781. The work of SG and MM was partly supported by NSF CCF-1535987, NSF IIS-1618662, and a Google Research Award. KS would like to acknowledge NSF CAREER Award 1750575 and Sloan Research Fellowship.

## Footnotes

[1] Note that the standard definition of Rademacher complexity assumes that hypothesis sets are not data-dependent, however the definition remains valid for data-dependent hypothesis sets.

[2]Sampling without replacement is only adopted to make the analysis more concise; its extension to sampling with replacement is straightforward.

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
