[Supplementary Material · hss-supp.pdf]

# A  Further background on stability

The study of stability dates back to early work on the analysis of $k$-neareast neighbor and other local discrimination rules [Rogers and Wagner, 1978, Devroye and Wagner, 1979]. Stability has been critically used in the analysis of stochastic optimization [Shalev-Shwartz et al., 2010] and online-to-batch conversion [Cesa-Bianchi et al., 2001]. Stability bounds have been generalized to the non-i.i.d. settings, including stationary [Mohri and Rostamizadeh, 2010] and non-stationary [Kuznetsov and Mohri, 2017] $\phi$-mixing and $\beta$-mixing processes. They have also been used to derive learning bounds for transductive inference [Cortes et al., 2008]. Stability bounds were further extended to cover *almost stable* algorithms by Kutin and Niyogi [2002]. These authors also discussed a number of alternative definitions of stability, see also [Kearns and Ron, 1997]. An alternative notion of stability was also used by Kale et al. [2011] to analyze $k$-fold cross-validation for a number of stable algorithms.

# B    Properties of data-dependent Rademacher complexity

In this section, we highlight several key properties of our notion of data-dependent Rademacher complexity.

## B.1    Upper-bound on Rademacher complexity of data-dependent hypothesis sets

**Lemma 2.** *For any sample $S = (x_1^S, \ldots, x_m^S) \in \mathbb{R}^N$, define the hypothesis set $\mathcal{H}_S$ as follows:*

$$\mathcal{H}_S = \left\{ x \mapsto w^S \cdot x \colon w^S = \sum_{i=1}^m \alpha_i x_i^S, \|\alpha\|_1 \leq \Lambda_1 \right\},$$

*where $\Lambda_1 \geq 0$. Define $r_T$ and $r_{S \cup T}$ as follows: $r_T = \sqrt{\frac{\sum_{i=1}^m \|x_i^T\|_2^2}{m}}$ and $r_{S \cup T} = \max_{x \in S \cup T} \|x\|_2$. Then, the empirical Rademacher complexity of the family of data-dependent hypothesis sets $\mathcal{H} = (\mathcal{H}_S)_{S \in \mathcal{X}^m}$ can be upper-bounded as follows:*

$$\widehat{\mathfrak{R}}_{S,T}^\diamond(\mathcal{H}) \leq r_T\, r_{S \cup T} \Lambda_1 \sqrt{\frac{2 \log(4m)}{m}} \leq r_{S \cup T}^2 \Lambda_1 \sqrt{\frac{2 \log(4m)}{m}}.$$

*Proof.*  The following inequalities hold:

$$\widehat{\mathfrak{R}}_{S,T}^\diamond(\mathcal{H}) = \frac{1}{m} \mathop{\mathbb{E}}_{\boldsymbol{\sigma}} \left[ \sup_{h \in \mathcal{H}_{S,T}^\sigma} \sum_{i=1}^m \sigma_i h(x_i^T) \right] = \frac{1}{m} \mathop{\mathbb{E}}_{\boldsymbol{\sigma}} \left[ \sup_{\|\alpha\|_1 \leq \Lambda_1} \sum_{i=1}^m \sigma_i \sum_{j=1}^m \alpha_j x_j^{S_T,\sigma} \cdot x_i^T \right]$$

$$= \frac{1}{m} \mathop{\mathbb{E}}_{\boldsymbol{\sigma}} \left[ \sup_{\|\alpha\|_1 \leq \Lambda_1} \sum_{j=1}^m \alpha_j \left( x_j^{S_T,\sigma} \sum_{i=1}^m \sigma_i \cdot x_i^T \right) \right]$$

$$= \frac{\Lambda_1}{m} \mathop{\mathbb{E}}_{\boldsymbol{\sigma}} \left[ \max_{j \in [m]} \left| x_j^{S_T,\sigma} \cdot \sum_{i=1}^m \sigma_i x_i^T \right| \right]$$

$$\leq \frac{\Lambda_1}{m} \mathop{\mathbb{E}}_{\boldsymbol{\sigma}} \left[ \max_{\substack{x' \in S \cup T \\ \sigma' \in \{-1,+1\}}} \sum_{i=1}^m \sigma_i (\sigma' x' \cdot x_i^T) \right].$$

The norm of the vector $z' \in \mathbb{R}^m$ with coordinates $(\sigma' x' \cdot x_i^T)$ can be bounded as follows:

$$\sqrt{\sum_{i=1}^m (\sigma' x' \cdot x_i^T)^2} \leq \|x'\| \sqrt{\sum_{i=1}^m \|x_i^T\|^2} \leq r_{S \cup T} \sqrt{m}\, r_T.$$

Thus, by Massart's lemma, since $|S \cup T| \leq 2m$, the following inequality holds:

$$\widehat{\mathfrak{R}}_{S,T}^\diamond(\mathcal{H}) \leq r_T\, r_{S \cup T} \Lambda_1 \sqrt{\frac{2 \log(4m)}{m}} \leq r_{S \cup T}^2 \Lambda_1 \sqrt{\frac{2 \log(4m)}{m}},$$

which completes the proof.    $\square$

Notice that the bound on the Rademacher complexity in Lemma 2 is non-trivial since it depends on the samples $S$ and $T$, while a standard Rademacher complexity for non-data-dependent hypothesis set containing $\mathcal{H}_S$ would require taking a maximum over all samples $S$ of size $m$.

**Lemma 3.** *Suppose $\mathcal{X} = \mathbb{R}^N$, and for every sample $S \in \mathcal{Z}^m$ we associate a matrix $A_S \in \mathbb{R}^{d \times N}$ for some $d > 0$, and let $\mathcal{W}_{S,\Lambda} = \{w \in \mathbb{R}^d \colon \|A_S^\top w\|_2 \leq \Lambda\}$ for some $\Lambda > 0$. Consider the hypothesis set $\mathcal{H}_S := \left\{ x \mapsto w^\top A_S x \colon w \in \mathcal{W}_{S,\Lambda} \right\}$. Then, the empirical Rademacher complexity of the family of data-dependent hypothesis sets $\mathcal{H} = (\mathcal{H}_S)_{S \in \mathcal{Z}^m}$ can be upper-bounded as follows:*

$$\widehat{\mathfrak{R}}_{S,T}^\diamond(\mathcal{H}) \leq \frac{\Lambda \sqrt{\sum_{i=1}^m \|x_i^T\|_2^2}}{m} \leq \frac{\Lambda r}{\sqrt{m}},$$

*where $r = \sup_{i \in [m]} \|x_i^T\|_2$.*

*Proof.* Let $X_T = [x_1^T \cdots x_m^T]$. The following inequalities hold:

$$\widehat{\mathfrak{R}}_{S,T}^{\diamond}(\mathcal{H}) = \frac{1}{m}\mathop{\mathbb{E}}_{\boldsymbol{\sigma}}\left[\sup_{h\in\mathcal{H}_{S,T}^{\sigma}}\sum_{i=1}^{m}\sigma_i h(x_i^T)\right] = \frac{1}{m}\mathop{\mathbb{E}}_{\boldsymbol{\sigma}}\left[\sup_{w:\,\|A_S^\top w\|_2\leq\Lambda} w^\top A_S X_T \boldsymbol{\sigma}\right]$$

$$\leq \frac{\Lambda}{m}\mathop{\mathbb{E}}_{\boldsymbol{\sigma}}\left[\|X_T\boldsymbol{\sigma}\|_2\right] \qquad\qquad \text{(Cauchy-Schwarz)}$$

$$\leq \frac{\Lambda}{m}\sqrt{\mathop{\mathbb{E}}_{\boldsymbol{\sigma}}\left[\|X_T\boldsymbol{\sigma}\|_2^2\right]} \qquad\qquad \text{(Jensen's ineq.)}$$

$$\leq \frac{\Lambda}{m}\sqrt{\mathop{\mathbb{E}}_{\boldsymbol{\sigma}}\left[\sum_{i,j=1}^{m}\sigma_i\sigma_j(x_i^T\cdot x_j^T)\right]}$$

$$= \frac{\Lambda\sqrt{\sum_{i=1}^{m}\|x_i^T\|_2^2}}{m},$$

which completes the proof. $\qquad\square$

## B.2   Concentration

**Lemma 4.** *Let $\mathcal{H}$ a family of $\beta$-stable data-dependent hypothesis sets. Then, for any $\delta > 0$, with probability at least $1-\delta$ (over the draw of two samples $S$ and $T$ with size $m$), the following inequality holds:*

$$\left|\widehat{\mathfrak{R}}_{S,T}^{\diamond}(\mathcal{G}) - \mathfrak{R}_m^{\diamond}(\mathcal{G})\right| \leq \sqrt{\frac{[(m\beta+1)^2 + m^2\beta^2]\log\frac{2}{\delta}}{2m}}.$$

*Proof.* Let $T'$ be a sample differing from $T$ only by point. Fix $\eta > 0$. For any $\boldsymbol{\sigma}$, by definition of the supremum, there exists $h' \in \mathcal{H}_{S,T'}^{\sigma}$ such that:

$$\sum_{i=1}^{m}\sigma_i L(h', z_i^T) \geq \sup_{h\in\mathcal{H}_{S,T'}^{\sigma}}\sum_{i=1}^{m}\sigma_i L(h, z_i^{T'}) - \eta.$$

By the $\beta$-stability of $\mathcal{H}$, there exists $h \in \mathcal{H}_{S,T}^{\sigma}$ such that for any $z \in \mathcal{Z}$, $|L(h', z) - L(h, z)| \leq \beta$. Thus, we have

$$\sup_{h\in\mathcal{H}_{S,T'}^{\sigma}}\sum_{i=1}^{m}\sigma_i L(h, z_i^{T'}) \leq \sum_{i=1}^{m}\sigma_i L(h', z_i^{T'}) + \eta \leq \sum_{i=1}^{m}[\sigma_i(L(h, z_i^{T'}) + \beta)] + \eta.$$

Since the inequality holds for all $\eta > 0$, we have

$$\frac{1}{m}\sup_{h\in\mathcal{H}_{S,T'}^{\sigma}}\sum_{i=1}^{m}\sigma_i L(h, z_i^{T'}) \leq \frac{1}{m}\sum_{i=1}^{m}\sigma_i(L(h, z_i^{T'}) + \beta) \leq \frac{1}{m}\sup_{h\in\mathcal{H}_{S,T}^{\sigma}}\sum_{i=1}^{m}\sigma_i L(h, z_i^T) + \beta + \frac{1}{m}.$$

Thus, replacing $T$ by $T'$ affects $\widehat{\mathfrak{R}}_{S,T}^{\diamond}(\mathcal{G})$ by at most $\beta + \frac{1}{m}$. By the same argument, changing sample $S$ by one point modifies $\widehat{\mathfrak{R}}_{S,T}^{\diamond}(\mathcal{G})$ at most by $\beta$. Thus, by McDiarmid's inequality, for any $\delta > 0$, with probability at least $1-\delta$, the following inequality holds:

$$\left|\widehat{\mathfrak{R}}_{S,T}^{\diamond}(\mathcal{G}) - \mathfrak{R}_m^{\diamond}(\mathcal{G})\right| \leq \sqrt{\frac{[(m\beta+1)^2 + m^2\beta^2]\log\frac{2}{\delta}}{2m}}.$$

This completes the proof. $\qquad\square$

## C Proof of Lemma 1

*Proof.* Let $S \in \mathcal{Z}^m$, $z \in S$, and $z' \in \mathcal{Z}$. For any $h \in \mathcal{H}_S$ and $h' \in \mathcal{H}_{S^{z \leftrightarrow z'}}$, by the $\beta$-uniform stability of $\mathcal{H}$, there exists $h'' \in \mathcal{H}_S$ such that $L(h', z) - L(h'', z) \leq \beta$. Thus,

$$L(h', z) - L(h, z) = L(h', z) - L(h'', z) + L(h'', z) - L(h, z) \leq \beta + \sup_{h'', h \in \mathcal{H}_S} L(h'', z) - L(h, z).$$

This implies the inequality

$$\sup_{h \in \mathcal{H}_S, h' \in \mathcal{H}_{S^{z \leftrightarrow z'}}} L(h', z) - L(h, z) \leq \beta + \sup_{h'', h \in \mathcal{H}_S} L(h'', z) - L(h, z),$$

and the lemma follows. $\square$

# D  Proof of Theorem 1

In this section, we present the proof of Theorem 1.

*Proof.* We will use the following symmetrization result, which holds for any $\epsilon > 0$ with $n\epsilon^2 \geq 2$ for data-dependent hypothesis sets (Lemma 5 below):

$$\mathbb{P}_{S \sim \mathcal{D}^m} \left[ \sup_{h \in \mathcal{H}_S} R(h) - \widehat{R}_S(h) > \epsilon \right] \leq 2 \mathbb{P}_{\substack{S \sim \mathcal{D}^m \\ T \sim \mathcal{D}^n}} \left[ \sup_{h \in \mathcal{H}_S} \widehat{R}_T(h) - \widehat{R}_S(h) > \frac{\epsilon}{2} \right]. \tag{13}$$

Thus, we will seek to bound the right-hand side as follows, where we write $(S, T) \sim U$ to indicate that the sample $S$ of size $m$ is drawn uniformly without replacement from $U$ and that $T$ is the remaining part of $U$, that is $(S, T) = U$:

$$\mathbb{P}_{\substack{S \sim \mathcal{D}^m \\ T \sim \mathcal{D}^n}} \left[ \sup_{h \in \mathcal{H}_S} \widehat{R}_T(h) - \widehat{R}_S(h) > \frac{\epsilon}{2} \right]$$

$$= \mathbb{E}_{U \sim \mathcal{D}^{m+n}} \left[ \mathbb{P}_{\substack{(S,T) \sim U \\ |S|=m, |T|=n}} \left[ \sup_{h \in \mathcal{H}_S} \widehat{R}_T(h) - \widehat{R}_S(h) > \frac{\epsilon}{2} \right] \Big| U \right]$$

$$\leq \mathbb{E}_{U \sim \mathcal{D}^{m+n}} \left[ \mathbb{P}_{\substack{(S,T) \sim U \\ |S|=m, |T|=n}} \left[ \sup_{h \in \overline{\mathcal{H}}_{U,m}} \widehat{R}_T(h) - \widehat{R}_S(h) > \frac{\epsilon}{2} \right] \Big| U \right].$$

To upper bound the probability inside the expectation, we use an extension of McDiarmid's inequality to sampling without replacement [Cortes et al., 2008], which applies to symmetric functions. We can apply that extension to $\Phi(S) = \sup_{h \in \overline{\mathcal{H}}_{U,m}} \widehat{R}_T(h) - \widehat{R}_S(h)$, for a fixed $U$, since $\Phi(S)$ is a symmetric function of the sample points $z_1, \ldots, z_m$) in $S$. Changing one point in $S$ affects $\Phi(S)$ at most by $\frac{1}{m} + \frac{1}{m} = \frac{m+u}{mu}$, thus, by the extension of McDiarmid's inequality to sampling without replacement, for a fixed $U \in \mathcal{Z}^{m+n}$, the following inequality holds:

$$\mathbb{P}_{\substack{(S,T) \sim U \\ |S|=m, |T|=n}} \left[ \sup_{h \in \overline{\mathcal{H}}_{U,m}} \widehat{R}_T(h) - \widehat{R}_S(h) > \frac{\epsilon}{2} \right] \leq \exp \left[ - \frac{2}{\eta} \frac{mn}{m+n} \left( \frac{\epsilon}{2} - \mathbb{E}[\Phi(S)] \right)^2 \right], \tag{14}$$

where $\eta = \frac{m+n}{m+n-\frac{1}{2}} \frac{1}{1 - \frac{1}{2\max\{m,n\}}} \leq 3$. Plugging in the bound on $\mathbb{E}[\Phi(S)]$ of Lemma 6 below, and setting

$$\epsilon = \max_{U \in \mathcal{Z}^{m+n}} 2\widehat{\mathfrak{R}}^{\diamond}_{U,m}(\mathcal{G}) + 3\sqrt{\left( \frac{1}{m} + \frac{1}{n} \right) \log(\frac{2}{\delta})} + 2\sqrt{\left( \frac{1}{m} + \frac{1}{n} \right)^3 mn},$$

which satisfies $n\epsilon^2 \geq 2$, it is easy to check that the RHS in (14) becomes smaller than $\frac{\delta}{2}$. This in turn implies, via (13), that the probability that $\sup_{h \in \mathcal{H}_S} R(h) - \widehat{R}_S(h) > \epsilon$ is at most $\delta$, completing the proof. □

The following lemma shows that the standard symmetrization lemma holds for data-dependent hypothesis sets. This observation was already made by Gat [2001] (see also Lemma 2 in [Cannon et al., 2002]) for the symmetrization lemma of Vapnik [1998][p. 139], used by the author in the case $n = m$. However, that symmetrization lemma of Vapnik [1998] holds only for random variables taking values in $\{0, 1\}$ and its proof is not complete since the hypergeometric inequality is not proven.

**Lemma 5.** *Let $n \geq 1$ and fix $\epsilon > 0$ such that $n\epsilon^2 \geq 2$. Then, the following inequality holds:*

$$\mathbb{P}_{S \sim \mathcal{D}^m} \left[ \sup_{h \in \mathcal{H}_S} R(h) - \widehat{R}_S(h) > \epsilon \right] \leq 2 \mathbb{P}_{\substack{S \sim \mathcal{D}^m \\ T \sim \mathcal{D}^n}} \left[ \sup_{h \in \mathcal{H}_S} \widehat{R}_T(h_S) - \widehat{R}_S(h_S) > \frac{\epsilon}{2} \right].$$

*Proof.* The proof is standard. Below, we are giving a concise version mainly for the purpose of verifying that the data-dependency of the hypothesis set does not affect its correctness.

Fix $\eta > 0$. By definition of the supremum, there exists $h_S \in \mathcal{H}_S$ such that

$$\sup_{h \in \mathcal{H}_S} R(h) - \widehat{R}_S(h) - \eta \leq R(h_S) - \widehat{R}_S(h_S).$$

Since $\widehat{R}_T(h_S) - \widehat{R}_S(h_S) = \widehat{R}_T(h_S) - R(h_S) + R(h_S) - \widehat{R}_S(h_S)$, we can write

$$1_{\widehat{R}_T(h_S) - \widehat{R}_S(h_S) > \frac{\epsilon}{2}} \geq 1_{\widehat{R}_T(h_S) - R(h_S) > -\frac{\epsilon}{2}} 1_{R(h_S) - \widehat{R}_S(h_S) > \epsilon} = 1_{R(h_S) - \widehat{R}_T(h_S) < \frac{\epsilon}{2}} 1_{R(h_S) - \widehat{R}_S(h_S) > \epsilon}.$$

Thus, for any $S \in \mathcal{Z}^m$, taking the expectation of both sides with respect to $T$ yields

$$\mathbb{P}_{T \sim \mathcal{D}^n}\left[\widehat{R}_T(h_S) - \widehat{R}_S(h_S) > \frac{\epsilon}{2}\right] \geq \mathbb{P}_{T \sim \mathcal{D}^n}\left[R(h_S) - \widehat{R}_T(h_S) < \frac{\epsilon}{2}\right] 1_{R(h_S) - \widehat{R}_S(h_S) > \epsilon}$$

$$= \left[1 - \mathbb{P}_{T \sim \mathcal{D}^n}\left[R(h_S) - \widehat{R}_T(h_S) \geq \frac{\epsilon}{2}\right]\right] 1_{R(h_S) - \widehat{R}_S(h_S) > \epsilon}$$

$$\leq \left[1 - \frac{4 \operatorname{Var}[L(h_S, z)]}{n\epsilon^2}\right] 1_{R(h_S) - \widehat{R}_S(h_S) > \epsilon} \qquad \text{(Chebyshev's ineq.)}$$

$$\geq \left[1 - \frac{1}{n\epsilon^2}\right] 1_{R(h_S) - \widehat{R}_S(h_S) > \epsilon},$$

where the last inequality holds since $L(h_S, z)$ takes values in $[0, 1]$:

$$\operatorname{Var}[L(h_S, z)] = \mathbb{E}_{z \sim \mathcal{D}}[L^2(h_S, z)] - \mathbb{E}_{z \sim \mathcal{D}}[L(h_S, z)]^2 \leq \mathbb{E}_{z \sim \mathcal{D}}[L(h_S, z)] - \mathbb{E}_{z \sim \mathcal{D}}[L(h_S, z)]^2$$

$$= \mathbb{E}_{z \sim \mathcal{D}}[L(h_S, z)](1 - \mathbb{E}_{z \sim \mathcal{D}}[L(h_S, z)]) \leq \frac{1}{4}.$$

Taking expectation with respect to $S$ gives

$$\mathbb{P}_{\substack{S \sim \mathcal{D}^m \\ T \sim \mathcal{D}^n}}\left[\widehat{R}_T(h_S) - \widehat{R}_S(h_S) > \frac{\epsilon}{2}\right] \geq \left[1 - \frac{1}{n\epsilon^2}\right] \mathbb{P}_{S \sim \mathcal{D}^m}\left[R(h_S) - \widehat{R}_S(h_S) > \epsilon\right]$$

$$\geq \frac{1}{2} \mathbb{P}_{S \sim \mathcal{D}^m}\left[R(h_S) - \widehat{R}_S(h_S) > \epsilon\right] \qquad (n\epsilon^2 \geq 2)$$

$$\geq \frac{1}{2} \mathbb{P}_{S \sim \mathcal{D}^m}\left[\sup_{h \in \mathcal{H}_S} R(h) - \widehat{R}_S(h) > \epsilon + \eta\right].$$

Since the inequality holds for all $\eta > 0$, by the right-continuity of the cumulative distribution function, it implies

$$\mathbb{P}_{\substack{S \sim \mathcal{D}^m \\ T \sim \mathcal{D}^n}}\left[\widehat{R}_T(h_S) - \widehat{R}_S(h_S) > \frac{\epsilon}{2}\right] \geq \frac{1}{2} \mathbb{P}_{S \sim \mathcal{D}^m}\left[\sup_{h \in \mathcal{H}_S} R(h) - \widehat{R}_S(h) > \epsilon\right].$$

Since $h_S$ is in $\mathcal{H}_S$, by definition of the supremum, we have

$$\mathbb{P}_{\substack{S \sim \mathcal{D}^m \\ T \sim \mathcal{D}^n}}\left[\sup_{h \in \mathcal{H}_S} \widehat{R}_T(h) - \widehat{R}_S(h) > \frac{\epsilon}{2}\right] \geq \mathbb{P}_{\substack{S \sim \mathcal{D}^m \\ T \sim \mathcal{D}^n}}\left[\widehat{R}_T(h_S) - \widehat{R}_S(h_S) > \frac{\epsilon}{2}\right],$$

which completes the proof. $\qquad\qquad\square$

The following lemma provides a bound on $\mathbb{E}[\Phi(S)]$:

**Lemma 6.** *Fix $U \in \mathcal{Z}^{m+n}$. Then, the following upper bound holds:*

$$\mathbb{E}_{\substack{(S,T) \sim U \\ |S|=m, |T|=n}}\left[\sup_{h \in \overline{\mathcal{H}}_{U,m}} \widehat{R}_T(h) - \widehat{R}_S(h)\right] \leq \widehat{\mathfrak{R}}_{U,m}^{\diamond}(\mathcal{G}) + \sqrt{\frac{\log(2e)(m+n)^3}{2(mn)^2}}.$$

*For $m = n$, the inequality becomes:*

$$\mathbb{E}_{\substack{(S,T) \sim U \\ |S|=m, |T|=n}}\left[\sup_{h \in \overline{\mathcal{H}}_{U,m}} \widehat{R}_T(h) - \widehat{R}_S(h)\right] \leq \widehat{\mathfrak{R}}_{U,m}^{\diamond}(\mathcal{G}) + 2\sqrt{\frac{\log(2e)}{m}}.$$

*Proof.* The proof is an extension of the analysis of *maximum discrepancy* in [Bartlett and Mendelson, 2002]. Let $|\boldsymbol{\sigma}|$ denote $\sum_{i=1}^{m+n} \sigma_i$ and let $I \subseteq \left[ -\frac{(m+n)^2}{m}, \frac{(m+n)^2}{n} \right]$ denote the set of values $|\boldsymbol{\sigma}|$ can take. For any $q \in I$, define $s(q)$ as follows:

$$s(q) = \mathbb{E}_{\boldsymbol{\sigma}} \left[ \sup_{h \in \overline{\mathcal{H}}_{U,m}} \frac{1}{m+n} \sum_{i=1}^{m+n} \sigma_i L(h, z_i^U) \middle| |\boldsymbol{\sigma}| = q \right].$$

Let $|\boldsymbol{\sigma}|_+$ denote the number of positive $\sigma_i$s, taking value $\frac{m+n}{n}$, then $|\boldsymbol{\sigma}|$ can be expressed as follows:

$$|\boldsymbol{\sigma}| = \sum_{i=1}^{m+n} \sigma_i = |\boldsymbol{\sigma}|_+ \frac{m+n}{n} - (m+n - |\boldsymbol{\sigma}|_+) \frac{m+n}{m} = \frac{(m+n)^2}{mn}(|\boldsymbol{\sigma}|_+ - n). \tag{15}$$

Thus, we have $|\boldsymbol{\sigma}| = 0$ iff $|\boldsymbol{\sigma}|_+ = m$, and the condition ($|\boldsymbol{\sigma}| = 0$) precisely corresponds to having the equality

$$\frac{1}{m+n} \sum_{i=1}^{m+n} \sigma_i L(h, z_i^U) = \widehat{R}_T(h) - \widehat{R}_S(h),$$

where $S$ is the sample of size $m$ defined by those $z_i$s for which $\sigma_i$ takes value $\frac{m+n}{n}$. In view of that, we have

$$\mathbb{E}_{\substack{(S,T) \sim U \\ |S|=m, |T|=n}} \left[ \sup_{h \in \overline{\mathcal{H}}_{U,m}} \widehat{R}_T(h) - \widehat{R}_S(h) \right] = s(0).$$

Let $q_1, q_2 \in I$, with $q_1 = p_1 \frac{m+n}{n} - (m+n-p_1) \frac{m+n}{m}$, $q_2 = p_2 \frac{m+n}{n} - (m+n-p_2) \frac{m+n}{m}$ and $q_1 \le q_2$. Then, we can write

$$s(q_1) = \mathbb{E} \left[ \sup_{g \in G} \sum_{i=1}^{p_1} \frac{1}{n} L(h, z_i) - \sum_{i=p_1+1}^{m+n} \frac{1}{m} L(h, z_i) \right]$$

$$s(q_2) = \mathbb{E} \left[ \sup_{g \in G} \sum_{i=1}^{p_1} \frac{1}{n} L(h, z_i) - \sum_{i=p_1+1}^{m+n} \frac{1}{m} L(h, z_i) + \sum_{i=p_1+1}^{p_2} \left[ \frac{1}{n} + \frac{1}{m} \right] L(h, z_i) \right].$$

Thus, we have the following Lipschitz property:

$$|s(q_2) - s(q_1)| \le |p_2 - p_1| \left[ \frac{1}{m} + \frac{1}{n} \right] = |(p_2 - n) - (p_1 - n)| \left[ \frac{1}{m} + \frac{1}{n} \right] \qquad \text{(using (15))}$$

$$= |q_2 - q_1| \frac{mn}{(m+n)^2} \left[ \frac{1}{m} + \frac{1}{n} \right]$$

$$= \frac{|q_2 - q_1|}{m+n}.$$

By this Lipschitz property, we can write

$$\mathbb{P} \left[ |s(|\boldsymbol{\sigma}|) - s(\mathbb{E}[|\boldsymbol{\sigma}|])| > \epsilon \right] \le \mathbb{P} \left[ ||\boldsymbol{\sigma}| - \mathbb{E}[|\boldsymbol{\sigma}|]| > (m+n)\epsilon \right] \le 2 \exp \left[ -2 \frac{(mn)^2 \epsilon^2}{(m+n)^3} \right],$$

since the range of each $\sigma_i$ is $\frac{m+n}{n} + \frac{m+n}{m} = \frac{(m+n)^2}{mn}$. We now use this inequality to bound the second moment of $Z = s(|\boldsymbol{\sigma}|) - s(\mathbb{E}[|\boldsymbol{\sigma}|]) = s(|\boldsymbol{\sigma}|) - s(0)$, as follows, for any $u \ge 0$:

$$\mathbb{E}[Z^2] = \int_0^{+\infty} \mathbb{P}[Z^2 > t] \, dt$$

$$= \int_0^u \mathbb{P}[Z^2 > t] \, dt + \int_u^{+\infty} \mathbb{P}[Z^2 > t] \, dt$$

$$\le u + 2 \int_u^{+\infty} \exp \left[ -2 \frac{(mn)^2 t}{(m+n)^3} \right] dt$$

$$\le u + \left[ \frac{(m+n)^3}{(mn)^2} \exp \left[ -2 \frac{(mn)^2 t}{(m+n)^3} \right] \right]_u^{+\infty}$$

$$= u + \frac{(m+n)^3}{(mn)^2} \exp \left[ -2 \frac{(mn)^2 u}{(m+n)^3} \right].$$

Choosing $u = \frac{1}{2} \frac{\log(2)(m+n)^3}{(mn)^2}$ to minimize the right-hand side gives $\mathbb{E}[Z^2] \leq \frac{\log(2e)(m+n)^3}{2(mn)^2}$. By Jensen's inequality, this implies $\mathbb{E}[|Z|] \leq \sqrt{\frac{\log(2e)(m+n)^3}{2(mn)^2}}$ and therefore

$$\underset{\substack{(S,T)\sim U \\ |S|=m,|T|=n}}{\mathbb{E}} \left[ \sup_{h \in \overline{\mathcal{H}}_{U,m}} \widehat{R}_T(h) - \widehat{R}_S(h) \right] = s(0) \leq \mathbb{E}[s(|\boldsymbol{\sigma}|)] + \sqrt{\frac{\log(2e)(m+n)^3}{2(mn)^2}}.$$

Since we have $\mathbb{E}[s(|\boldsymbol{\sigma}|)] = \widehat{\mathfrak{R}}^{\diamond}_{U,m}(\mathcal{G})$, this completes the proof. $\qquad\square$

# E Proof of Theorem 2

In this section, we present the full proof of Theorem 2. The proof of each of the three bounds (9), (10) and (11) are given in separate subsections.

## E.1 Proof of bound (9)

*Proof.* For any two samples $S, S'$, define the $\Psi(S, S')$ as follows:

$$\Psi(S, S') = \sup_{h \in \mathcal{H}_S} R(h) - \widehat{R}_{S'}(h).$$

The proof consists of applying McDiarmid's inequality to $\Psi(S, S)$. For any sample $S'$ differing from $S$ by one point, we can decompose $\Psi(S, S) - \Psi(S', S')$ as follows:

$$\Psi(S, S) - \Psi(S', S') = \big[\Psi(S, S) - \Psi(S, S')\big] + \big[\Psi(S, S') - \Psi(S', S')\big].$$

Now, by the sub-additivity of the $\sup$ operation, the first term can be upper-bounded as follows:

$$\Psi(S, S) - \Psi(S, S') \leq \sup_{h \in \mathcal{H}_S} \big[R(h) - \widehat{R}_S(h)\big] - \big[R(h) - \widehat{R}_{S'}(h)\big]$$

$$\leq \sup_{h \in \mathcal{H}_S} \frac{1}{m}\big[L(h, z) - L(h, z')\big] \leq \frac{1}{m},$$

where we denoted by $z$ and $z'$ the labeled points differing in $S$ and $S'$ and used the 1-boundedness of the loss function.

We now analyze the second term:

$$\Psi(S, S') - \Psi(S', S') = \sup_{h \in \mathcal{H}_S} \big[R(h) - \widehat{R}_{S'}(h)\big] - \sup_{h \in \mathcal{H}_{S'}} \big[R(h) - \widehat{R}_{S'}(h)\big].$$

By definition of the supremum, for any $\epsilon > 0$, there exists $h \in \mathcal{H}_S$ such that

$$\sup_{h \in \mathcal{H}_S} \big[R(h) - \widehat{R}_{S'}(h)\big] - \epsilon \leq \big[R(h) - \widehat{R}_{S'}(h)\big]$$

By the $\beta$-stability of $(H_S)_{S \in \mathcal{Z}^m}$, there exists $h' \in \mathcal{H}_{S'}$ such that for all $z$, $|L(h, z) - L(h', z)| \leq \beta$. In view of these inequalities, we can write

$$\Psi(S, S') - \Psi(S', S') \leq \big[R(h) - \widehat{R}_{S'}(h)\big] + \epsilon - \sup_{h \in \mathcal{H}_{S'}} \big[R(h) - \widehat{R}_{S'}(h)\big]$$

$$\leq \big[R(h) - \widehat{R}_{S'}(h)\big] + \epsilon - \big[R(h') - \widehat{R}_{S'}(h')\big]$$

$$\leq \big[R(h) - R(h')\big] + \epsilon + \big[\widehat{R}_{S'}(h') - \widehat{R}_{S'}(h)\big]$$

$$\leq \epsilon + 2\beta.$$

Since the inequality holds for any $\epsilon > 0$, it implies that $\Psi(S, S') - \Psi(S', S') \leq 2\beta$. Summing up the bounds on the two terms shows the following:

$$\Psi(S, S) - \Psi(S', S') \leq \frac{1}{m} + 2\beta.$$

Thus, by McDiarmid's inequality, for any $\delta > 0$, with probability at least $1 - \delta$, we have

$$\Psi(S, S) \leq \mathbb{E}\big[\Psi(S, S)\big] + (1 + 2\beta m)\sqrt{\tfrac{1}{2m} \log(\tfrac{1}{\delta})}. \tag{16}$$

We now bound $\mathbb{E}[\Psi(S,S)]$ by $2\mathfrak{R}_m^\diamond(\mathcal{G})$ as follows:

$$\underset{S\sim\mathcal{D}^m}{\mathbb{E}}\big[\Psi(S,S)\big]$$

$$= \underset{S\sim\mathcal{D}^m}{\mathbb{E}}\left[\sup_{h\in\mathcal{H}_S}\big[R(h)-\widehat{R}_S(h)\big]\right]$$

$$= \underset{S\sim\mathcal{D}^m}{\mathbb{E}}\left[\sup_{h\in\mathcal{H}_S}\left[\underset{T\sim\mathcal{D}^m}{\mathbb{E}}\big[\widehat{R}_T(h)\big]-\widehat{R}_S(h)\right]\right] \qquad \text{(def. of } R(h))$$

$$\leq \underset{S,T\sim\mathcal{D}^m}{\mathbb{E}}\left[\sup_{h\in\mathcal{H}_S}\widehat{R}_T(h)-\widehat{R}_S(h)\right] \qquad \text{(sub-additivity of sup)}$$

$$= \underset{S,T\sim\mathcal{D}^m}{\mathbb{E}}\left[\sup_{h\in\mathcal{H}_S}\frac{1}{m}\sum_{i=1}^m\Big[L(h,z_i^T)-L(h,z_i^S)\Big]\right]$$

$$= \underset{S,T\sim\mathcal{D}^m}{\mathbb{E}}\left[\underset{\boldsymbol{\sigma}}{\mathbb{E}}\left[\sup_{h\in\mathcal{H}_{S,T}^{\boldsymbol{\sigma}}}\frac{1}{m}\sum_{i=1}^m\sigma_i\Big[L(h,z_i^T)-L(h,z_i^S)\Big]\right]\right] \qquad \text{(symmetry)}$$

$$\leq \underset{\substack{S,T\sim\mathcal{D}^m\\\boldsymbol{\sigma}}}{\mathbb{E}}\left[\sup_{h\in\mathcal{H}_{S,T}^{\boldsymbol{\sigma}}}\frac{1}{m}\sum_{i=1}^m\sigma_iL(h,z_i^T)+\sup_{h\in\mathcal{H}_{S,T}^{\boldsymbol{\sigma}}}\frac{1}{m}\sum_{i=1}^m-\sigma_iL(h,z_i^S)\right] \qquad \text{(sub-additivity of sup)}$$

$$= \underset{\substack{S,T\sim\mathcal{D}^m\\\boldsymbol{\sigma}}}{\mathbb{E}}\left[\sup_{h\in\mathcal{H}_{S,T}^{\boldsymbol{\sigma}}}\frac{1}{m}\sum_{i=1}^m\sigma_iL(h,z_i^T)+\sup_{h\in\mathcal{H}_{T,S}^{-\boldsymbol{\sigma}}}\frac{1}{m}\sum_{i=1}^m-\sigma_iL(h,z_i^S)\right] \qquad (\mathcal{H}_{S,T}^{\boldsymbol{\sigma}}=\mathcal{H}_{T,S}^{-\boldsymbol{\sigma}})$$

$$= \underset{\substack{S,T\sim\mathcal{D}^m\\\boldsymbol{\sigma}}}{\mathbb{E}}\left[\sup_{h\in\mathcal{H}_{S,T}^{\boldsymbol{\sigma}}}\frac{1}{m}\sum_{i=1}^m\sigma_iL(h,z_i^T)+\sup_{h\in\mathcal{H}_{T,S}^{\boldsymbol{\sigma}}}\frac{1}{m}\sum_{i=1}^m\sigma_iL(h,z_i^S)\right] \qquad \text{(symmetry)}$$

$$= 2\mathfrak{R}_m^\diamond(\mathcal{G}). \qquad \text{(linearity of expectation)}$$

Now, we show that $\mathbb{E}_{S\sim\mathcal{D}^m}[\Psi(S,S)]\leq\bar{\chi}$. To do so, first fix $\epsilon>0$. By definition of the supremum, for any $S\in\mathcal{Z}^m$, there exists $h_S$ such that the following inequality holds:

$$\sup_{h\in\mathcal{H}_S}\big[R(h)-\widehat{R}_S(h)\big]-\epsilon\leq R(h_S)-\widehat{R}_S(h_S).$$

Now, by definition of $R(h_S)$, we can write

$$\underset{S\sim\mathcal{D}^m}{\mathbb{E}}\big[R(h_S)\big]=\underset{S\sim\mathcal{D}^m}{\mathbb{E}}\left[\underset{z\sim\mathcal{D}}{\mathbb{E}}(L(h_S,z))\right]=\underset{\substack{S\sim\mathcal{D}^m\\z\sim\mathcal{D}}}{\mathbb{E}}\big[L(h_S,z)\big].$$

Then, by the linearity of expectation, we can also write

$$\underset{S\sim\mathcal{D}^m}{\mathbb{E}}\big[\widehat{R}_S(h_S)\big]=\underset{\substack{S\sim\mathcal{D}^m\\z\sim S}}{\mathbb{E}}\big[L(h_S,z)\big]=\underset{\substack{S\sim\mathcal{D}^m\\z'\sim\mathcal{D}\\z\sim S}}{\mathbb{E}}\big[L(h_{S^{z\leftrightarrow z'}},z')\big].$$

In view of these two equalities, we can now rewrite the upper bound as follows:

$$\underset{S\sim\mathcal{D}^m}{\mathbb{E}}\big[\Psi(S,S)\big]\leq\underset{S\sim\mathcal{D}^m}{\mathbb{E}}\big[R(h_S)-\widehat{R}_S(h_S)\big]+\epsilon$$

$$= \underset{\substack{S\sim\mathcal{D}^m\\z'\sim\mathcal{D}}}{\mathbb{E}}\big[L(h_S,z')\big]-\underset{\substack{S\sim\mathcal{D}^m\\z'\sim\mathcal{D}\\z\sim S}}{\mathbb{E}}\big[L(h_{S^{z\leftrightarrow z'}},z')\big]+\epsilon$$

$$= \underset{\substack{S\sim\mathcal{D}^m\\z'\sim\mathcal{D}\\z\sim S}}{\mathbb{E}}\big[L(h_S,z')-L(h_{S^{z\leftrightarrow z'}},z')\big]+\epsilon$$

$$= \underset{\substack{S\sim\mathcal{D}^m\\z'\sim\mathcal{D}\\z\sim S}}{\mathbb{E}}\big[L(h_{S^{z\leftrightarrow z'}},z)-L(h_S,z)\big]+\epsilon$$

$$\leq \bar{\chi}+\epsilon.$$

Since the inequality holds for all $\epsilon>0$, it implies $\mathbb{E}_{S\sim\mathcal{D}^m}\big[\Psi(S,S)\big]\leq\bar{\chi}$. Plugging in these upper bounds on the expectation in the inequality (16) completes the proof. $\qquad\square$

### E.2 Proof of bound (10)

The proof of bound (10) relies on recent techniques introduced in the differential privacy literature to derive improved generalization guarantees for stable data-dependent hypothesis sets [Steinke and Ullman, 2017, Bassily et al., 2016] (see also [McSherry and Talwar, 2007]). Our proof also benefits from the recent improved stability results of Feldman and Vondrak [2018]. We will make use of the following lemma due to Steinke and Ullman [2017, Lemma 1.2], which reduces the task of deriving a concentration inequality to that of upper bounding an expectation of a maximum.

**Lemma 7.** *Fix $p \geq 1$. Let $X$ be a random variable with probability distribution $\mathcal{D}$ and $X_1, \ldots, X_p$ independent copies of $X$. Then, the following inequality holds:*

$$\mathop{\mathbb{P}}_{X \sim \mathcal{D}} \left[ X \geq 2 \mathop{\mathbb{E}}_{X_k \sim \mathcal{D}} \left[ \max \left\{ 0, X_1, \ldots, X_p \right\} \right] \right] \leq \frac{\log 2}{p}.$$

We will also use the following result which, under a sensitivity assumption, further reduces the task of upper bounding the expectation of the maximum to that of bounding a more favorable expression.

**Lemma 8** ([McSherry and Talwar, 2007, Bassily et al., 2016, Feldman and Vondrak, 2018]). *Let $f_1, \ldots, f_p \colon \mathcal{Z}^m \to \mathbb{R}$ be $p$ functions with sensitivity $\Delta$. Let $\mathcal{A}$ be the algorithm that, given a dataset $S \in \mathcal{Z}^m$ and a parameter $\epsilon > 0$, returns the index $k \in [p]$ with probability proportional to $e^{\frac{\epsilon f_k(S)}{2\Delta}}$. Then, $\mathcal{A}$ is $\epsilon$-differentially private and, for any $S \in \mathcal{Z}^m$, the following inequality holds:*

$$\max_{k \in [p]} \left\{ f_k(S) \right\} \leq \mathop{\mathbb{E}}_{k = \mathcal{A}(S)} \left[ f_k(S) \right] + \frac{2\Delta}{\epsilon} \log p.$$

Notice that, if we define $f_{p+1} = 0$, then, by the same result, the algorithm $\mathcal{A}$ returning the index $k \in [p+1]$ with probability proportional to $e^{\frac{\epsilon f_k(S) \mathbf{1}_{k \neq (p+1)}}{2\Delta}}$ is $\epsilon$-differentially private and the following inequality holds for any $S \in \mathcal{Z}^m$:

$$\max \left\{ 0, \max_{k \in [p]} \left\{ f_k(S) \right\} \right\} = \max_{k \in [p+1]} \left\{ f_k(S) \right\} \leq \mathop{\mathbb{E}}_{k = \mathcal{A}(S)} \left[ f_k(S) \right] + \frac{2\Delta}{\epsilon} \log(p+1). \tag{17}$$

Equipped with these lemmas, we can now turn to the proof of bound (10):

*Proof.* For any two samples $S, S'$ of size $m$, define $\Psi(S, S')$ as follows:

$$\Psi(S, S') = \sup_{h \in \mathcal{H}_S} R(h) - \widehat{R}_{S'}(h).$$

The proof consists of deriving a high-probability bound for $\Psi(S, S)$. To do so, by Lemma 7 applied to the random variable $X = \Psi(S, S)$, it suffices to bound $\mathbb{E}_{\mathsf{S} \sim \mathcal{D}^{pm}} \left[ \max \left\{ 0, \max_{k \in [p]} \left\{ \Psi(S_k, S_k) \right\} \right\} \right]$, where $\mathsf{S} = (S_1, \ldots, S_p)$ with $S_k$, $k \in [p]$, independent samples of size $m$ drawn from $\mathcal{D}^m$.

To bound that expectation, we can use Lemma 8 and instead bound $\mathbb{E}_{\substack{\mathsf{S} \sim \mathcal{D}^{pm} \\ k = \mathcal{A}(\mathsf{S})}} [\Psi(S_k, S_k)]$, where $\mathcal{A}$ is an $\epsilon$-differentially private algorithm.

Now, to apply Lemma 8, we first show that, for any $k \in [p]$, the function $f_k \colon \mathsf{S} \to \Psi(S_k, S_k)$ is $\Delta$-sensitive with $\Delta = \frac{1}{m} + 2\beta$. Fix $k \in [p]$. Let $\mathsf{S}' = (S'_1, \ldots, S'_p)$ be in $\mathcal{Z}^{pm}$ and assume that $\mathsf{S}'$ differs from $\mathsf{S}$ by one point. If they differ by a point not in $S_k$ (or $S'_k$), then $f_k(\mathsf{S}) = f_k(\mathsf{S}')$. Otherwise, they differ only by a point in $S_k$ (or $S'_k$) and $f_k(\mathsf{S}) - f_k(\mathsf{S}') = \Psi(S_k, S_k) - \Psi(S'_k, S'_k)$. We can decompose this term as follows:

$$\Psi(S_k, S_k) - \Psi(S'_k, S'_k) = \left[ \Psi(S_k, S_k) - \Psi(S_k, S'_k) \right] + \left[ \Psi(S_k, S'_k) - \Psi(S'_k, S'_k) \right].$$

Now, by the sub-additivity of the sup operation, the first term can be upper-bounded as follows:

$$\Psi(S_k, S_k) - \Psi(S_k, S'_k) \leq \sup_{h \in \mathcal{H}_{S_k}} \left[ R(h) - \widehat{R}_{S_k}(h) \right] - \left[ R(h) - \widehat{R}_{S'_k}(h) \right]$$

$$\leq \sup_{h \in \mathcal{H}_{S_k}} \frac{1}{m} \left[ L(h, z) - L(h, z') \right] \leq \frac{1}{m},$$

where we denoted by $z$ and $z'$ the labeled points differing in $S_k$ and $S'_k$ and used the 1-boundedness of the loss function.

We now analyze the second term:

$$\Psi(S_k, S'_k) - \Psi(S'_k, S'_k) = \sup_{h \in \mathcal{H}_{S_k}} \left[ R(h) - \widehat{R}_{S'_k}(h) \right] - \sup_{h \in \mathcal{H}_{S'_k}} \left[ R(h) - \widehat{R}_{S'_k}(h) \right].$$

By definition of the supremum, for any $\eta > 0$, there exists $h \in \mathcal{H}_{S_k}$ such that

$$\sup_{h \in \mathcal{H}_{S_k}} \left[ R(h) - \widehat{R}_{S'_k}(h) \right] - \eta \le \left[ R(h) - \widehat{R}_{S'_k}(h) \right]$$

By the $\beta$-stability of $(\mathcal{H}_S)_{S \in \mathcal{Z}^m}$, there exists $h' \in \mathcal{H}_{S'_k}$ such that for all $z$, $\left| L(h, z) - L(h', z) \right| \le \beta$. In view of these inequalities, we can write

$$\Psi(S_k, S'_k) - \Psi(S'_k, S'_k) \le \left[ R(h) - \widehat{R}_{S'_k}(h) \right] + \eta - \sup_{h \in \mathcal{H}_{S'_k}} \left[ R(h) - \widehat{R}_{S'_k}(h) \right]$$

$$\le \left[ R(h) - \widehat{R}_{S'_k}(h) \right] + \eta - \left[ R(h') - \widehat{R}_{S'_k}(h') \right]$$

$$\le \left[ R(h) - R(h') \right] + \eta + \left[ \widehat{R}_{S'_k}(h') - \widehat{R}_{S'_k}(h) \right]$$

$$\le \eta + 2\beta.$$

Since the inequality holds for any $\eta > 0$, it implies that $\Psi(S_k, S'_k) - \Psi(S'_k, S'_k) \le 2\beta$. Summing up the bounds on the two terms shows the following:

$$\Psi(S_k, S_k) - \Psi(S'_k, S'_k) \le \frac{1}{m} + 2\beta.$$

Having established the $\Delta$-sensitivity of the functions $f_k$, $k \in [p]$, we can now apply Lemma 8. Fix $\epsilon > 0$. Then, by Lemma 8 and (17), the algorithm $\mathcal{A}$ returning $k \in [p+1]$ with probability proportional to $e^{\frac{\epsilon \Psi(S_k, S_k) 1_{k \ne (p+1)}}{2\Delta}}$ is $\epsilon$-differentially private and, for any sample $\mathsf{S} \in \mathcal{Z}^{pm}$, the following inequality holds:

$$\max \left\{ 0, \max_{k \in [p]} \left\{ \Psi(S_k, S_k) \right\} \right\} \le \underset{k = \mathcal{A}(\mathsf{S})}{\mathbb{E}} \left[ \Psi(S_k, S_k) \right] + \frac{2\Delta}{\epsilon} \log(p+1).$$

Taking the expectation of both sides yields

$$\underset{\mathsf{S} \sim \mathcal{D}^{pm}}{\mathbb{E}} \left[ \max \left\{ 0, \max_{k \in [p]} \left\{ \Psi(S_k, S_k) \right\} \right\} \right] \le \underset{\substack{\mathsf{S} \sim \mathcal{D}^{pm} \\ k = \mathcal{A}(\mathsf{S})}}{\mathbb{E}} \left[ \Psi(S_k, S_k) \right] + \frac{2\Delta}{\epsilon} \log(p+1). \qquad (18)$$

We will show the following upper bound on the expectation: $\mathbb{E}_{\substack{\mathsf{S} \sim \mathcal{D}^{pm} \\ k = \mathcal{A}(\mathsf{S})}} \left[ \Psi(S_k, S_k) \right] \le (e^\epsilon - 1) + e^\epsilon \chi$.

To do so, first fix $\eta > 0$. By definition of the supremum, for any $S \in \mathcal{Z}^m$, there exists $h_S \in \mathcal{H}_S$ such that the following inequality holds:

$$\sup_{h \in \mathcal{H}_S} \left[ R(h) - \widehat{R}_S(h) \right] - \eta \le R(h_S) - \widehat{R}_S(h_S).$$

In what follows, we denote by $\mathsf{S}^{k, z \leftrightarrow z'} \in \mathcal{Z}^{pm}$ the result of modifying $\mathsf{S} = (S_1, \ldots, S_p) \in \mathcal{Z}^{pm}$ by replacing $z \in S_k$ with $z'$.

Now, by definition of the algorithm $\mathcal{A}$, we can write:

$$\underset{\substack{S\sim\mathcal{D}^{pm}\\k=\mathcal{A}(S)}}{\mathbb{E}}\left[R(h_{S_k})\right] = \underset{\substack{S\sim\mathcal{D}^{pm}\\k=\mathcal{A}(S)}}{\mathbb{E}}\left[\underset{z'\sim\mathcal{D}}{\mathbb{E}}\left[L(h_{S_k},z')\right]\right] \qquad \text{(def. of } R(h_{S_k}))$$

$$= \underset{\substack{S\sim\mathcal{D}^{pm}\\z'\sim\mathcal{D}}}{\mathbb{E}}\left[\sum_{k=1}^{p}\mathbb{P}[\mathcal{A}(S)=k]\,L(h_{S_k},z')\right] \qquad \text{(def. of } \underset{k=\mathcal{A}(S)}{\mathbb{E}})$$

$$= \sum_{k=1}^{p}\underset{\substack{S\sim\mathcal{D}^{pm}\\z'\sim\mathcal{D}}}{\mathbb{E}}\left[\mathbb{P}[\mathcal{A}(S)=k]\,L(h_{S_k},z')\right] \qquad \text{(linearity of expect.)}$$

$$\le \sum_{k=1}^{p}\underset{\substack{S\sim\mathcal{D}^{pm}\\z'\sim\mathcal{D},\,z\sim S_k}}{\mathbb{E}}\left[e^{\epsilon}\,\mathbb{P}[\mathcal{A}(S^{k,z\leftrightarrow z'})=k]\,L(h_{S_k},z')\right] \quad \text{($\epsilon$-diff. privacy of $\mathcal{A}$)}$$

$$= \sum_{k=1}^{p}\underset{\substack{S\sim\mathcal{D}^{pm}\\z'\sim\mathcal{D},\,z\sim S_k}}{\mathbb{E}}\left[e^{\epsilon}\,\mathbb{P}[\mathcal{A}(S)=k]\,L(h_{S_k^{z\leftrightarrow z'}},z)\right] \qquad \text{(swapping $z'$ and $z$)}$$

$$\le \sum_{k=1}^{p}\underset{\substack{S\sim\mathcal{D}^{pm}\\z'\sim\mathcal{D},\,z\sim S_k}}{\mathbb{E}}\left[e^{\epsilon}\,\mathbb{P}[\mathcal{A}(S)=k]\,L(h_{S_k},z)\right]+e^{\epsilon}\chi. \quad \text{(By Lemma 9 below)}$$

Now, observe that $\mathbb{E}_{z\sim S_k}\left[L(h_{S_k},z)\right]$ coincides with $\widehat{R}(h_{S_k})$, the empirical loss of $h_{S_k}$. Thus, we can write

$$\underset{\substack{S\sim\mathcal{D}^{pm}\\k=\mathcal{A}(S)}}{\mathbb{E}}\left[R(h_{S_k})\right] \le \sum_{k=1}^{p}\underset{\substack{S\sim\mathcal{D}^{pm}\\z\sim S_k}}{\mathbb{E}}\left[e^{\epsilon}\,\mathbb{P}[\mathcal{A}(S)=k]\,\widehat{R}_{S_k}(h_{S_k})\right]+e^{\epsilon}\chi,$$

and therefore

$$\underset{\substack{S\sim\mathcal{D}^{pm}\\k=\mathcal{A}(S)}}{\mathbb{E}}\left[\Psi(S_k,S_k)\right] \le \sum_{k=1}^{p}\underset{\substack{S\sim\mathcal{D}^{pm}\\k=\mathcal{A}(S)}}{\mathbb{E}}\left[(e^{\epsilon}-1)\widehat{R}_{S_k}(h_{S_k})\right]+e^{\epsilon}\chi+\eta$$

$$\le (e^{\epsilon}-1)+e^{\epsilon}\chi+\eta.$$

Since the inequality holds for any $\eta>0$, we have

$$\underset{\substack{S\sim\mathcal{D}^{pm}\\k=\mathcal{A}(S)}}{\mathbb{E}}\left[\Psi(S_k,S_k)\right] \le (e^{\epsilon}-1)+e^{\epsilon}\chi.$$

Thus, by (18), the following inequality holds:

$$\underset{S\sim\mathcal{D}^{pm}}{\mathbb{E}}\left[\max\left\{0,\max_{k\in[p]}\left\{\Psi(S_k,S_k)\right\}\right\}\right] \le (e^{\epsilon}-1)+e^{\epsilon}\chi+\frac{2\Delta}{\epsilon}\log(p+1). \qquad (19)$$

For any $\delta\in(0,1)$, choose $p=\frac{\log 2}{\delta}$, which implies $\log(p+1)=\log\left[\frac{2+\delta}{\delta}\right]\le\log\frac{3}{\delta}$. Then, by Lemma 7, with probability at least $1-\delta$ over the draw of a sample $S\sim\mathcal{D}^m$, the following inequality holds for all $h\in\mathcal{H}_S$:

$$R(h) \le \widehat{R}_S(h)+(e^{\epsilon}-1)+e^{\epsilon}\chi+\frac{2\Delta}{\epsilon}\log\left[\frac{3}{\delta}\right]. \qquad (20)$$

For $\epsilon\le\frac{1}{2}$, the inequality $(e^{\epsilon}-1)\le 2\epsilon$ holds. Thus,

$$(e^{\epsilon}-1)+e^{\epsilon}\chi+\frac{2\Delta}{\epsilon}\log\left[\frac{3}{\delta}\right] \le 2\epsilon+\sqrt{e}\chi+\frac{2\Delta}{\epsilon}\log\left[\frac{3}{\delta}\right]$$

Choosing $\epsilon=\sqrt{\Delta\log\left[\frac{3}{\delta}\right]}$ gives

$$R(h) \le \widehat{R}_S(h)+\sqrt{e}\chi+4\sqrt{\Delta\log\left[\frac{3}{\delta}\right]}$$

$$= \widehat{R}_S(h)+\sqrt{e}\chi+4\sqrt{\left[\frac{1}{m}+2\beta\right]\log\left[\frac{3}{\delta}\right]}.$$

Combining this inequality with the inequality of Theorem 2 related to the Rademacher complexity:

$$\forall h \in \mathcal{H}_S, R(h) \leq \widehat{R}_S(h) + 2\mathfrak{R}_m^\diamond(\mathcal{G}) + [1 + 2\beta m]\sqrt{\frac{\log\frac{1}{\delta}}{2m}}, \tag{21}$$

and using the union bound complete the proof. $\square$

The following is a helper lemma for the analysis in the above proof:

**Lemma 9.** *The following upper bound in terms of the CV-stability coefficient $\chi$ holds:*

$$\sum_{k=1}^{p} \mathop{\mathbb{E}}_{\substack{S \sim \mathcal{D}^{pm} \\ z' \sim \mathcal{D}, \, z \sim S_k}} \left[ e^\epsilon \, \mathbb{P}[\mathcal{A}(S) = k] \left[ L(h_{S_k^{z \leftrightarrow z'}}, z) - L(h_{S_k}, z) \right] \right] \leq e^\epsilon \chi.$$

*Proof.* Upper bounding the difference of losses by a supremum to make the CV-stability coefficient appear gives the following chain of inequalities:

$$\sum_{k=1}^{p} \mathop{\mathbb{E}}_{\substack{S \sim \mathcal{D}^{pm} \\ z' \sim \mathcal{D}, \, z \sim S_k}} \left[ e^\epsilon \, \mathbb{P}[\mathcal{A}(S) = k] \left[ L(h_{S_k^{z \leftrightarrow z'}}, z) - L(h_{S_k}, z) \right] \right]$$

$$\leq \sum_{k=1}^{p} \mathop{\mathbb{E}}_{\substack{S \sim \mathcal{D}^{pm} \\ z' \sim \mathcal{D}, \, z \sim S_k}} \left[ e^\epsilon \, \mathbb{P}[\mathcal{A}(S) = k] \sup_{h \in \mathcal{H}_{S_k}, \, h' \in \mathcal{H}_{S_k^{z \leftrightarrow z'}}} [L(h', z) - L(h, z)] \right]$$

$$= \sum_{k=1}^{p} \mathop{\mathbb{E}}_{S \sim \mathcal{D}^{pm}} \left[ e^\epsilon \, \mathbb{P}[\mathcal{A}(S) = k] \mathop{\mathbb{E}}_{z' \sim \mathcal{D}, \, z \sim S_k} \left[ \sup_{h \in \mathcal{H}_{S_k}, \, h' \in \mathcal{H}_{S_k^{z \leftrightarrow z'}}} [L(h', z) - L(h, z)] \mid S \right] \right]$$

$$\leq \sum_{k=1}^{p} \mathop{\mathbb{E}}_{S \sim \mathcal{D}^{pm}} \left[ e^\epsilon \, \mathbb{P}[\mathcal{A}(S) = k] \chi \right]$$

$$= \mathop{\mathbb{E}}_{S \sim \mathcal{D}^{pm}} \left[ \sum_{k=1}^{p} \mathbb{P}[\mathcal{A}(S) = k] \right] \cdot e^\epsilon \chi$$

$$= e^\epsilon \chi,$$

which completes the proof. $\square$

### E.3 Proof of bound (11)

Bound (11) is a simple consequence of the fact that the composition of the two stages of the learning algorithm is uniformly-stable in the classical sense. Specifically, consider a learning algorithm that consists of determining the hypothesis set $\mathcal{H}_S$ based on the sample $S$ and then selecting an arbitrary (but fixed) hypothesis $h_S \in \mathcal{H}_S$. The following lemma shows that the uniform-stability coefficient of this learning algorithm can be bounded in terms of its hypothesis set stability and its max-diameter.

**Lemma 10.** *Suppose the family of data-dependent hypothesis sets $\mathcal{H} = (\mathcal{H}_S)_{S \in \mathcal{Z}^m}$ is $\beta$-uniformly stable and admits max-diameter $\Delta_{\max}$. Then, for any two samples $S, S' \in \mathcal{Z}^m$ differing in exactly one point, and for any $z \in \mathcal{Z}$, we have*

$$|L(h_S, z) - L(h_{S'}, z)| \leq 3\beta + \Delta_{\max}.$$

*Proof.* We first show that for any two hypotheses $h, h' \in \mathcal{H}_S$ and for any $z \in \mathcal{Z}$, we have $|L(h, z) - L(h', z)| \leq 2\beta + \Delta_{\max}$. Indeed, let $S''$ be a sample obtained by replacing an arbitrary point in $S$ by $z$. Then, by $\beta$-uniform hypothesis set stability of $\mathcal{H}$, there exist hypotheses $g, g' \in \mathcal{H}_{S''}$ such that $|L(h, z) - L(g, z)| \leq \beta$ and $|L(h', z) - L(g', z)| \leq \beta$. Furthermore, since $z \in S''$, we have $|L(g, z) - L(g', z)| \leq \Delta_{\max}$. By combining these inequalities, we get that $|L(h, z) - L(h', z)| \leq 2\beta + \Delta_{\max}$, as required.

Now, let $h' \in \mathcal{H}_S$ be a hypothesis such that $|L(h', z) - L(h_{S'}, z)| \leq \beta$. Since $h', h_S \in \mathcal{H}_S$, by the analysis in the preceding paragraph, we have $|L(h_S, z) - L(h', z)| \leq 2\beta + \Delta_{\max}$. Combining these two inequalities, we have $|L(h_S, z) - L(h_{S'}, z)| \leq 3\beta + \Delta_{\max}$, completing the proof. $\square$

Finally, bound (11) follows immediately from the following result of Feldman and Vondrak [2019], setting $\ell(S, z) := L(h_S, z)$ and $\gamma = 3\beta + \Delta_{\max}$, and the fact that any two hypotheses $h$ and $h'$ in $\mathcal{H}_S$ differ in loss on any point $z$ by at most $\Delta_{\max}$ in order to get a bound which holds uniformly for all $h \in \mathcal{H}_S$.

**Theorem 3** ([Feldman and Vondrak, 2019]). *Let $\ell \colon \mathcal{Z}^m \times \mathcal{Z} \to [0,1]$ be a data-dependent function with uniform stability $\gamma$, i.e. for any $S, S' \in \mathcal{Z}^m$ differing in one point, and any $z \in \mathcal{Z}$, we have $|\ell(S, z) - \ell(S', z)| \leq \gamma$. Then, for any $\delta > 0$, with probability at least $1 - \delta$ over the choice of the sample S, the following inequality holds:*

$$\left| \mathop{\mathbb{E}}_{z \sim \mathcal{D}}[\ell(S, z)] - \mathop{\mathbb{E}}_{z \sim S}[\ell(S, z)] \right| \leq 47\gamma \log(m) \log\left(\tfrac{5m^3}{\delta}\right) + \sqrt{\tfrac{4}{m} \log\left(\tfrac{4}{\delta}\right)}.$$

# F  Extensions

We briefly discuss here some extensions of the framework and results presented in the previous section.

## F.1  Almost everywhere hypothesis set stability

As for standard algorithmic uniform stability, our generalization bounds for hypothesis set stability can be extended to the case where hypothesis set stability holds only with high probability [Kutin and Niyogi, 2002].

**Definition 4.** *Fix $m \geq 1$. We will say that a family of data-dependent hypothesis sets $\mathcal{H} = (\mathcal{H}_S)_{S \in \mathcal{Z}^m}$ is* weakly $(\beta, \delta)$-stable *for some $\beta \geq 0$ and $\delta > 0$, if, with probability at least $1 - \delta$ over the draw of a sample $S \in \mathcal{Z}^m$, for any sample $S'$ of size $m$ differing from $S$ only by one point, the following holds:*

$$\forall h \in \mathcal{H}_S, \exists h' \in \mathcal{H}_{S'}: \forall z \in \mathcal{Z}, |L(h, z) - L(h', z)| \leq \beta. \tag{22}$$

Notice that, in this definition, $\beta$ and $\delta$ depend on the sample size $m$. In practice, we often have $\beta = O(\frac{1}{m})$ and $\delta = O(e^{-\Omega(m)})$. The learning bounds of Theorem 2 can be straightforwardly extended to guarantees for weakly $(\beta, \delta)$-stable families of data-dependent hypothesis sets, by using a union bound and the confidence parameter $\delta$.

## F.2  Randomized algorithms

The generalization bounds given in this paper assume that the data-dependent hypothesis set $\mathcal{H}_S$ is *deterministic* conditioned on $S$. However, in some applications such as bagging, it is more natural to think of $\mathcal{H}_S$ as being constructed by a *randomized* algorithm with access to an independent source of randomness in the form of a random seed $s$. Our generalization bounds can be extended in a straightforward manner for this setting if the following can be shown to hold: there is a *good* set of seeds, $G$, such that (a) $\mathbb{P}[s \in G] \geq 1 - \delta$, where $\delta$ is the confidence parameter, and (b) conditioned on any $s \in G$, the family of data-dependent hypothesis sets $\mathcal{H} = (\mathcal{H}_S)_{S \in \mathcal{Z}^m}$ is $\beta$-uniformly stable. In that case, for any good set $s \in G$, Theorem 2 holds. Then taking a union bound, we conclude that with probability at least $1 - 2\delta$ over both the choice of the random seed $s$ and the sample set $S$, the generalization bounds hold. This can be further combined with almost-everywhere hypothesis stability as in section F.1 via another union bound if necessary.

## F.3  Data-dependent priors

An alternative scenario extending our study is one where, in the first stage, instead of selecting a hypothesis set $\mathcal{H}_S$, the learner decides on a probability distribution $\mathsf{p}_S$ on a fixed family of hypotheses $\mathcal{H}$. The second stage consists of using that *prior* $\mathsf{p}_S$ to choose a hypothesis $h_S \in \mathcal{H}$, either deterministically or via a randomized algorithm. Our notion of hypothesis set stability could then be extended to that of stability of priors and lead to new learning bounds depending on that stability parameter. This could lead to data-dependent prior bounds somewhat similar to the PAC-Bayesian bounds [Catoni, 2007, Parrado-Hernández et al., 2012, Lever et al., 2013, Dziugaite and Roy, 2018a,b], but with technically quite different guarantees.

# G   Other applications

## G.1   Anti-distillation

A similar setup to distillation (section 5.4) is that of *anti-distillation* where the predictor $f_S^*$ in the first stage is chosen from a simpler family, say that of linear hypotheses, and where the sample-dependent hypothesis set $\mathcal{H}_S$ is the subset of a very rich family $\mathcal{H}$. $\mathcal{H}_S$ is defined as the set of predictors that are close to $f_S^*$:

$$\mathcal{H}_S = \left\{ h \in \mathcal{H} \colon (\| (h - f_S^*) \|_\infty \leq \gamma) \wedge (\| (h - f_S^*) 1_S \|_\infty \leq \Delta) \right\},$$

with $\Delta = O(1/\sqrt{m})$. Thus, the restriction to $S$ of a hypothesis $h \in \mathcal{H}_S$ is close to $f_S^*$ in $\ell_\infty$-norm. As shown in section 5.4, the family of hypothesis sets $\mathcal{H}_S$ is $\mu\beta$-stable. However, here, the hypothesis sets $\mathcal{H}_S$ could be very complex and the Rademacher complexity $\mathfrak{R}_m^\diamond(\mathcal{H})$ not very favorable. Nevertheless, by Theorem 2, for any $\delta > 0$, with probability at least $1 - \delta$ over the draw of a sample $S \sim \mathcal{D}^m$, the following inequality holds for any $h \in \mathcal{H}_S$:

$$R(h) \leq \widehat{R}_S(h) + \sqrt{e}\mu(\Delta + \beta) + 4\sqrt{(\tfrac{1}{m} + 2\mu\beta) \log(\tfrac{6}{\delta})}.$$

Notice that a standard uniform-stability does not apply here since the $(1/\sqrt{m})$-closeness of the hypotheses to $f_S^*$ on $S$ does not imply their global $(1/\sqrt{m})$-closeness.

## G.2   Principal Components Regression

Principal Components Regression is a very commonly used technique in data analysis. In this setting, $\mathcal{X} \subseteq \mathbb{R}^d$ and $\mathcal{Y} \subseteq \mathbb{R}$, with a loss function $\ell$ that is $\mu$-Lipschitz in the prediction. Given a sample $S = \{(x_i, y_i) \in \mathcal{X} \times \mathcal{Y} \colon i \in [m]\}$, we learn a linear regressor on the data projected on the principal $k$-dimensional space of the data. Specifically, let $\Pi_S \in \mathbb{R}^{d \times d}$ be the projection matrix giving the projection of $\mathbb{R}^d$ onto the principal $k$-dimensional subspace of the data, i.e. the subspace spanned by the top $k$ left singular vectors of the design matrix $X_S = [x_1, x_2, \cdots, x_m]$. The hypothesis space $\mathcal{H}_S$ is then defined as $\mathcal{H}_S = \{x \mapsto w^\top \Pi_S x \colon w \in \mathbb{R}^k, \|w\| \leq \gamma\}$, where $\gamma$ is a predefined bound on the norm of the weight vector for the linear regressor. Thus, this can be seen as an instance of the setting in section 5.3, where the feature mapping $\Phi_S$ is defined as $\Phi_S(x) = \Pi_S x$.

To prove generalization bounds for this setup, we need to show that these feature mappings are stable. To do that, we make the following assumptions:

1. For all $x \in \mathcal{X}$, $\|x\| \leq r$ for some constant $r \geq 1$.
2. The data covariance matrix $\mathbb{E}_x[xx^\top]$ has a gap of $\lambda > 0$ between the $k$-th and $(k+1)$-th largest eigenvalues.

The matrix concentration bound of Rudelson and Vershynin [2007] implies that with probability at least $1 - \delta$ over the choice of $S$, we have $\|X_S X_S^\top - m \mathbb{E}_x[xx^\top]\| \leq cr^2\sqrt{m \log(m) \log(\tfrac{2}{\delta})}$ for some constant $c > 0$. Suppose $m$ is large enough so that $cr^2\sqrt{m \log(m) \log(\tfrac{2}{\delta})} \leq \tfrac{\lambda}{2}m$. Then, the gap between the $k$-th and $(k+1)$-th largest eigenvalues of $X_S X_S^\top$ is at least $\tfrac{\lambda}{2}m$. Now, consider changing one sample point $(x, y) \in S$ to $(x, y')$ to produce the sample set $S'$. Then, we have $X_{S'} X_{S'}^\top = X_S X_S^\top - xx^\top + x'x'^\top$. Since $\|-xx^\top + x'x'^\top\| \leq 2r^2$, by standard matrix perturbation theory bounds [Stewart, 1998], we have $\|\Pi_S - \Pi_{S'}\| \leq O(\tfrac{r^2}{\lambda m})$. Thus, $\|\Phi_S(x) - \Phi_{S'}(x)\| \leq \|\Pi_S - \Pi_{S'}\| \|x\| \leq O(\tfrac{r^3}{\lambda m})$.

Now, to apply the bound of (12), we need to compute a suitable bound on $\mathfrak{R}_m^\diamond(\mathcal{H})$. For this, we apply Lemma 3. For any $\|w\| \leq \gamma$, since $\|\Pi_S\| = 1$, we have $\|\Pi_S w\| \leq \gamma$. So the hypothesis set $\mathcal{H}_S' = \{x \mapsto w^\top \Pi_S x \colon w \in \mathbb{R}^k, \|\Pi_S w\| \leq \gamma\}$ contains $\mathcal{H}_S$. By Lemma 3, we have $\mathfrak{R}_m^\diamond(\mathcal{H}') \leq \tfrac{\gamma r}{\sqrt{m}}$. Thus, by plugging the bounds obtained above in (12), we conclude that with probability at least $1 - 2\delta$ over the choice of $S$, for any $h \in \mathcal{H}_S$, we have

$$R(h) \leq \widehat{R}_S(h) + O\left( \mu\gamma \frac{r^3}{\lambda} \sqrt{\frac{\log \tfrac{1}{\delta}}{m}} \right).$$

# H  PAC-Bayesian Bounds

The PAC-Bayes framework assumes a prior distribution $P$ over $\mathcal{H}$ and a posterior distribution $Q$ selected after observing the training sample. The framework helps derive learning bounds for randomized algorithms with probability distribution $Q$, in terms of the relative entropy of $Q$ and $P$.

In this section, we briefly discuss PAC-Bayesian learning bounds and present some key results. In Subsection H.1, we give PAC-Bayes learning bounds derived from Rademacher complexity bounds, which improve upon standard PAC-Bayes bounds [McAllester, 2003]. Similar bounds were already shown by Kakade et al. [2008] using elegant proofs based on strong convexity. Here, we give an alternative proof not invoking strong convexity. In Subsection H.2, we extend the PAC-Bayes framework to one where the prior distribution is selected after observing $S$ and will denote by $P_S$ that prior. Finally, in Subsection H.3, we briefly discuss derandomized PAC-Bayesian bounds, that is learning bounds derived for deterministic algorithms, using PAC-Bayes bounds.

## H.1  PAC-Bayes bounds derived from Rademacher complexity bounds

We will denote by $L_z$ the vector $(L(h, z))_{h \in \mathcal{H}}$. The expected loss of the randomized classifier $Q$ can then be written as $\mathbb{E}_{\substack{h \sim Q \\ z \sim \mathcal{D}}}[L(h, z)] = \mathbb{E}_{z \sim \mathcal{D}}[\langle Q, L_z \rangle]$.

Define $\mathcal{G}_\mu$ via $\mathcal{G}_\mu = \{Q \in \Delta(\mathcal{H}) : \mathsf{D}(Q \parallel P) \le \mu\}$, that is the family of distributions $Q$ defined over $\mathcal{H}$ with $\mu$-bounded relative entropy with respect to $P$. Then, by the standard Rademacher complexity bound [Koltchinskii and Panchenko, 2002, Mohri et al., 2018], for any $\delta > 0$, with probability at least $1 - \delta$ over the draw of a sample $S$ of size $m$, the following holds for all $Q \in \mathcal{G}_\mu$:

$$\mathbb{E}_{z \sim \mathcal{D}}[\langle Q, L_z \rangle] \le \mathbb{E}_{z \sim S}[\langle Q, L_z \rangle] + 2\mathfrak{R}_m(\mathcal{G}_\mu) + \sqrt{\frac{\log \frac{1}{\delta}}{2m}}. \tag{23}$$

We now give an upper bound on $\mathfrak{R}_m(\mathcal{G}_\mu)$. For any $Q$, define $\Psi(Q)$ by $\Psi(Q) = \mathsf{D}(Q, P)$ if $Q \in \Delta(\mathcal{H})$ and $+\infty$ otherwise. It is known that the conjugate function $\Psi^*$ of $\Psi$ is given by $\Psi^*(U) = \log\left(\mathbb{E}_{h \in P}[e^{U(h)}]\right)$, for all $U \in \mathbb{R}^{\mathcal{H}}$ (see for example [Mohri et al., 2018, Lemma B.37]). Let $U_{\boldsymbol{\sigma}} = \sum_{i=1}^m \sigma_i L_{z_i}$. Then, for any $t > 0$, we can write:

$$
\begin{aligned}
\mathfrak{R}_m(\mathcal{G}_\mu) &= \frac{1}{m} \mathbb{E}_{S, \boldsymbol{\sigma}}\left[\sup_{\mathsf{D}(Q \parallel P) \le \mu} \sum_{i=1}^m \sigma_i \langle Q, L_{z_i} \rangle\right] \\
&= \frac{1}{m} \mathbb{E}_{S, \boldsymbol{\sigma}}\left[\sup_{\mathsf{D}(Q \parallel P) \le \mu} \langle Q, U_{\boldsymbol{\sigma}} \rangle\right] && \text{(definition of } U_{\boldsymbol{\sigma}}) \\
&= \frac{1}{mt} \mathbb{E}_{S, \boldsymbol{\sigma}}\left[\sup_{\mathsf{D}(Q \parallel P) \le \mu} \langle Q, tU_{\boldsymbol{\sigma}} \rangle\right] && (t > 0) \\
&\le \frac{1}{mt} \mathbb{E}_{S, \boldsymbol{\sigma}}\left[\sup_{\Psi(Q) \le \mu} \Psi(Q) + \Psi^*(tU_{\boldsymbol{\sigma}})\right] && \text{(Fenchel inequality)} \\
&\le \frac{\mu}{mt} + \frac{1}{mt} \mathbb{E}_{S, \boldsymbol{\sigma}}[\Psi^*(tU_{\boldsymbol{\sigma}})].
\end{aligned}
$$

Now, we use the expression of $\Psi^*$ to bound the second term as follows:

$$
\begin{aligned}
\mathbb{E}_{S, \boldsymbol{\sigma}}\left[\Psi^*(tU_{\boldsymbol{\sigma}})\right] &= \mathbb{E}_{S, \boldsymbol{\sigma}}\left[\log\left(\mathbb{E}_{h \sim P}\left[e^{t \sum_{i=1}^m \sigma_i L(h, z_i)}\right]\right)\right] \\
&\le \mathbb{E}_S\left[\log\left(\mathbb{E}_{\boldsymbol{\sigma}, h \sim P}\left[e^{t \sum_{i=1}^m \sigma_i L(h, z_i)}\right]\right)\right] && \text{(Jensen's inequality)} \\
&= \mathbb{E}_S\left[\log\left(\mathbb{E}_{h \sim P}\left[\prod_{i=1}^m \cosh(tL(h, z_i))\right]\right)\right] \\
&\le \mathbb{E}_S\left[\log\left(\mathbb{E}_{h \sim P}\left[e^{m\frac{t^2}{2}}\right]\right)\right] = \frac{mt^2}{2}.
\end{aligned}
$$

Choosing $t = \sqrt{\frac{2\mu}{m}}$ to minimize the bound on the Rademacher complexity gives $\mathfrak{R}_m(\mathcal{G}_\mu) \leq \sqrt{\frac{2\mu}{m}}$. In view of that, (23) implies:

$$\mathbb{E}_{z \sim \mathcal{D}}\left[\langle Q, L_z \rangle\right] \leq \mathbb{E}_{z \sim S}\left[\langle Q, L_z \rangle\right] + 2\sqrt{\frac{2\mu}{m}} + \sqrt{\frac{\log \frac{1}{\delta}}{2m}}. \tag{24}$$

Proceeding as in [Kakade et al., 2008], by the union bound, the result can be extended to hold for any distribution $Q$, which is directly leading to the following result.

**Theorem 4.** *Let $P$ be a fixed prior on $\mathcal{H}$. Then, for any $\delta > 0$, with probability at least $1 - \delta$ over the draw of a sample $S$ of size $m$, the following holds for any posterior distribution $Q$ over $\mathcal{H}$:*

$$\mathbb{E}_{\substack{h \sim Q \\ z \sim \mathcal{D}}}\left[L(h, z)\right] \leq \mathbb{E}_{h \sim Q}\left[\frac{1}{m}\sum_{i=1}^{m} L(h, z_i)\right] + \left(4 + \frac{1}{\sqrt{e}}\right)\sqrt{\frac{\max\{\mathsf{D}(Q \parallel P), 1\}}{m}} + \sqrt{\frac{\log \frac{1}{\delta}}{2m}}.$$

This bound improves upon standard PAC-Bayes bounds (see for example [McAllester, 2003]) since it does not include an additive term in $\sqrt{(\log m)/m}$, as pointed by Kakade et al. [2008].

## H.2 Data-dependent PAC-Bayes bounds

In this section, we extend the framework to one where the prior distribution is selected after observing $S$ and will denote by $P_S$ that prior. To analyze that scenario, we can both use the general data-dependent learning bounds of Section 3, or the hypothesis set stability bounds of Section 4. We will focus here on the latter.

Define the data-dependent hypothesis set $\mathcal{G}_{S,\mu} = \{Q \in \Delta(\mathcal{H}) : \mathsf{D}(Q \parallel P_S) \leq \mu\}$ and assume that the priors $P_S$ are chosen so that $\mathcal{G}_\mu = (\mathcal{G}_{S,\mu})_S$ is $\beta$-stable. This may be by choosing $P_S$ and $P_{S'}$ to be close in total variation or relative entropy for any two samples $S$ and $S'$ differing by one point. Then, by Theorem 2, for any $\delta > 0$, with probably at least $1 - \delta$, the following holds for all $Q \in \mathcal{G}_{\mu,S}$:

$$\mathbb{E}_{\substack{h \sim Q \\ z \sim \mathcal{D}}}\left[L(h, z)\right] \leq \mathbb{E}_{h \sim Q}\left[\frac{1}{m}\sum_{i=1}^{m} L(h, z_i)\right] + \min\Bigg\{\min\left\{2\mathfrak{R}_m^\diamond(\mathcal{G}_\mu), \beta + \bar{\Delta}\right\} + (1 + 2\beta m)\sqrt{\frac{1}{2m}\log(\frac{1}{\delta})},$$

$$\sqrt{e}(\beta + \Delta) + 4\sqrt{(\frac{1}{m} + 2\beta)\log(\frac{6}{\delta})},$$

$$48(3\beta + \Delta_{\max})\log(m)\log(\frac{5m^3}{\delta}) + \sqrt{\frac{4}{m}\log(\frac{4}{\delta})}\Bigg\}.$$

The analysis of the Rademacher complexity $\mathfrak{R}_m^\diamond(\mathcal{G}_\mu)$ depends on the specific properties of the family of priors $P_S$. Here, we initiate its analysis and leave it to the reader to complete it for a choice of the priors.

Proceeding as in Subsection H.1, we define $\Psi_S$ by $\Psi_S(Q) = \mathsf{D}(Q, P_S)$ for any $Q \in \Delta(\mathcal{H})$ and denote by $\Psi_S^*$ its conjugate function. Let $U_{\boldsymbol{\sigma}} = \sum_{i=1}^{m} \sigma_i L_{z_i^T}$. Then, for any $t > 0$, we can write:

$$\mathfrak{R}_m^\diamond(\mathcal{G}_\mu) = \frac{1}{m}\mathbb{E}_{S,T,\boldsymbol{\sigma}}\left[\sup_{\mathsf{D}(Q \parallel P_{S_T^{\boldsymbol{\sigma}}}) \leq \mu}\sum_{i=1}^{m}\sigma_i\langle Q, L_{z_i}\rangle\right]$$

$$= \frac{1}{m}\mathbb{E}_{S,T,\boldsymbol{\sigma}}\left[\sup_{\mathsf{D}(Q \parallel P_{S_T^{\boldsymbol{\sigma}}}) \leq \mu}\langle Q, U_{\boldsymbol{\sigma}}\rangle\right] \qquad \text{(definition of } U_{\boldsymbol{\sigma}})$$

$$= \frac{1}{mt}\mathbb{E}_{S,T,\boldsymbol{\sigma}}\left[\sup_{\mathsf{D}(Q \parallel P_{S_T^{\boldsymbol{\sigma}}}) \leq \mu}\langle Q, tU_{\boldsymbol{\sigma}}\rangle\right] \qquad (t > 0)$$

$$\leq \frac{1}{mt}\mathbb{E}_{S,T,\boldsymbol{\sigma}}\left[\sup_{\Psi_{S_T^{\boldsymbol{\sigma}}}(Q) \leq \mu}\Psi_{S_T^{\boldsymbol{\sigma}}}(Q) + \Psi_{S_T^{\boldsymbol{\sigma}}}^*(tU_{\boldsymbol{\sigma}})\right] \qquad \text{(Fenchel inequality)}$$

$$\leq \frac{\mu}{mt} + \frac{1}{mt}\mathbb{E}_{S,T,\boldsymbol{\sigma}}\left[\Psi_{S_T^{\boldsymbol{\sigma}}}^*(tU_{\boldsymbol{\sigma}})\right].$$

Using the expression of the conjugate function $\Psi^*_{S_T^{\boldsymbol{\sigma}}}$, as in Subsection H.1, the second term can be bounded as follows:

$$\mathbb{E}_{S,T,\boldsymbol{\sigma}}\left[\Psi^*_{S_T^{\boldsymbol{\sigma}}}(tU_{\boldsymbol{\sigma}})\right] = \mathbb{E}_{S,T,\boldsymbol{\sigma}}\left[\log\left(\mathbb{E}_{h\sim P_{S_T^{\boldsymbol{\sigma}}}}\left[e^{t\sum_{i=1}^m \sigma_i L(h,z_i)}\right]\right)\right]$$

$$\leq \mathbb{E}_{S,T}\left[\log\left(\mathbb{E}_{\boldsymbol{\sigma},h\sim P_{S_T^{\boldsymbol{\sigma}}}}\left[e^{t\sum_{i=1}^m \sigma_i L(h,z_i)}\right]\right)\right] \qquad \text{(Jensen's inequality)}.$$

This last term can be bounded using Hoeffding's inequality and the specific properties of the priors leading to an explicit bound on the Rademacher complexity as in Subsection H.1.

## H.3 Derandomized PAC-Bayesian bounds

Derandomized versions of PAC-Bayesian bounds have been given in the past: margin bounds for linear predictors by McAllester [2003], more complex margin bounds by Neyshabur et al. [2018] where linear predictors are replaced with neural networks and where the norm-bound is replaced with a more complex norm condition, and chaining-based bounds by Miyaguchi [2019].

However, the benefit of these bounds is not clear since finer Rademacher complexity bounds can be derived for deterministic predictors. In fact, Rademacher complexity bounds can be used to derive finer PAC-Bayes bounds than existing ones. This was already pointed out by Kakade et al. [2008] and further shown here with an alternative proof and more favorable constants (Subsection H.1).

In fact, using the technique of obtaining KL-divergence between prior and posterior as upper bound on the Rademacher complexity, along with the optimistic rates in [Srebro et al., 2010], one can obtain just as in the previous section, an optimistic rate with data-dependent prior when one considers a non-negative smooth loss and, as predictor, the expected model under the posterior. As this is a straightforward application of the result of Srebro et al. [2010] combined with techniques presented here, we leave this for the reader to verify by themselves.