[Reviews · NeurIPS 2019]

Reviewer 1



The article is clearly written presenting high-quality research with original results. I have two main criticisms: 1) Beyond the pattern A lot of the examples analyzed seem to follow a similar pattern: the algorithm that selects some data-dependent “base” hypothesis is uniformly stable, and then the predictor returned is some linear combination of these base hypothesis. Is that an essential construction which allows for the hypothesis set stability analysis? 2) Related work comparison to PAC Bayes The introduction discusses related work, including to data-dependent priors. I think the text here misrepresents slightly the nature of this work, first by lumping all these contributions together, and second, by suggesting that the PAC Bayes framework applies only to randomized classifiers. In both cases, the text muddies the relationship with Dziugaite and Roy, who also exploit differential privacy results to obtain bounds for data-dependent hypothesis classes: to see this, note that their bounds, when viewed from the Radmacher perspective of Kakade, Sridharan, and Tewari (2009), are working with (nonuniform bounds over) data-dependent hypothesis sets. In more detail, the authors cite a number of works, including Catoni (2007), Parrado-Hernandez et al. (2012), Lever et al. (2013) and Dziugaite and Roy (2018a and 2018b). In fact, almost all of these are examples of distribution-dependent but data-independent priors: Catoni, Parrado et al. and Lever specify distribution-dependent priors and then BOUND the KL term to derive PAC-Bayes bounds that are in terms of known computable quantities. In Catoni and Lever these bounds are distribution independent (but note that a data-dependent choice of the inverse temperature term can makes the bound data-dependent). Parrado et al give two bounds: The first “data-dependent” prior is not data dependent in the sense meant here: the empirical risk calculation EXCLUDES the same data used in the prior. Their second bound, however, uses data and concentration of measure (about the mean) to derive a data-dependent BOUND on the KL term. This allows them to use a distribution-dependent hypothesis class, which is related to the ideas here. In contrast, Dziugaite and Roy study DATA-dependent priors and then employ differential privacy techniques, like those used here, to derive valid statistical bounds based on PAC-Bayes bounds. Further, they then show how weaker-than-private mechanisms yield valid bounds. They give a particular application using a type of Wasserstein control. The DP underpinnings of their PAC Bayes bounds and those derived here deserves to be highlighted. More generous and specific references would likely be appreciated. Second, the idea that PAC-Bayes bounds apply only to randomized classifiers is somewhat naive and further misrepresents the extent to which data-dependent priors bear on the same problem studied here. “Derandomization” techniques used in PAC-Bayes are well known. The original application of PAC Bayes to the SVM solution (a single hypothesis) is the classical example of obtaining bounds on a single hypothesis using PAC Bayes bounds and margin. These ideas have been used recently by various authors, including Neyshabur et al 2018 (ICLR) and Nagarajan and Kolter (2018). Typos: Line 140: \bar H_m The word “sensitivity” seems to be reused multiple times in the examples. The authors might want to consider defining more officially. Is it any different from bounded differences? Line 236: The authors mention that their bound for bagging is “similar (but incomparable) to the one derived by Elisseeff et al. [2005]”. Could the authors expand a bit? Line 3 typo “results” *** Update after Author Response: Thank you for pointing out the Distillation Example. It still assumes that the predictors in the high complexity hypothesis class satisfy bounded differences (and the loss is Lipchitz, so it seems that one essentially gets stability again). Then the output hypothesis is selected from an L-inf ball around the complex data-dependent hypothesis. I don't think that this qualifies as a "breaking the pattern" example. However, assuming author's claim "standard uniform-stability argument would not necessarily apply here" is correct, this example is still interesting. I was also wondering whether the assumption made in line 294 (that f_S-f_S' is in H) is a standard one. In summary, I am not convinced that this theory is yet useful for a larger range of problems or is superior to some other data-dependent analysis. However, I think it is theoretically intriguing and further developments of it could become high-impact and thus should be accepted. I am hoping that the authors will implement promised changes (expanding on PAC-Bayes connections, comparing to Elisseeff et al. bounds, and being upfront about the limitations of current applications).

Reviewer 2



The theoretical results are novel and sound. The authors show several application problems for their theoretical framework. I encourage the authors to discuss a bit regarding impact/audience during the rebuttal phase. === AFTER REBUTTAL I am satisfied with the authors response.

Reviewer 3



This paper is clearly written and the results are interesting and mathematically sound. The unification of the complexity-based and stability-based analysis into learning with data-dependent hypothesis set seems a significant contribution. Under the assumption of data-dependent stability, the generalization ability was proved in Theorem 2. The bound in Theorem 2 was obtained by combining several theoretical techniques. Then, Theorem 2 applies to a wide range of learning algorithms. minor comments: - I guess that the invariance under the sample permutation is implicitly assumed for the correspondence from S to H_S, i.e., the permutation of the sample S does not affect H_S. The last equation on page 3 uses such a condition. Supplementary comment on the invariance might be helpful for readers. - line 117: The sentence "hat{R}_T(H_{S,T}) is the standard empirical Rademacher complexity" sounds slightly confusing, since H_{S,T} also depends on the sample T. In my understanding, the standard Rademacher complexity does not deal with such a hypothesis set. - The term "sensitive" was separately defined in Section 5.2 and 5.4 but not in Section 5.3. It may be good to show a general definition of the sensitiveness before Section 5.2. ----- after Author Response: Thanks for the response. All points in my comments are now clear.

[Author Response · NeurIPS 2019]

We thank the reviewers for their constructive comments. Responses to the questions raised follow.

**Reviewer #2:**

1. **Going beyond the pattern:** While most of our applications indeed follow the pattern described by the reviewer, it is certainly not necessary for hypothesis set stability analysis. The distillation application in section 5.4 is one example which does not follow the pattern. Similarly, one can modify the application in section 5.3 by setting $\mathcal{H}_S$ to be any $\gamma$-Lipschitz function class (i.e. not necessarily a linear class as currently written), and the bounds follow verbatim. Similarly, the algorithm that selects the base hypotheses does not need to be uniformly stable: for example, in the bagging application of section 5.1, we get non-trivial generalization guarantees even if the base algorithm is *not* uniformly stable, as noted in line 235 of the paper.

2. **Related work comparison to PAC-Bayes bounds.** Thank you for elucidating the nuances of the prior work on PAC-Bayes bounds. We will certainly elaborate more on the points you raised in the next version of the paper.

3. **Typo corrections:** Thank you, we will fix the typos in the next version. We will also define sensitivity – it is indeed essentially the same as bounded differences. We will also add more details on the comparison of our bagging bounds with those of Elisseeff et al (2005).

**Reviewer #4:**

1. **Impact:** The impact of this paper can be judged from the contributions listed in section 1.1; additionally, we believe that our paper provides foundational work on analyzing generalization in data-dependent hypothesis sets.

2. **Audience:** We expect that any theoretically-minded ML researcher working with data-dependent hypothesis sets would find our paper interesting, and engineers may be able to use insights from our paper to design learning algorithms with good generalization properties. We therefore expect our paper to appeal to a large audience.

**Reviewer #6:**

1. **Invariance under sample permutation:** We indeed implicitly assume that $\mathcal{H}_S$ is invariant to permutation of samples. We will mention this explicitly in the next version.

2. $\hat{\mathfrak{R}}_T(\mathcal{H}_{S,T})$**:** Although the standard definition of empirical Rademacher complexity is for hypothesis sets that are not data-dependent, the definition still remains valid for data-dependent hypothesis sets. In particular, $\hat{\mathfrak{R}}_T(\mathcal{H}_{S,T})$ indeed coincides with the empirical Rademacher complexity of the hypothesis set $\mathcal{G}$ for the empirical sample $T$, with $\mathcal{G} = \mathcal{H}_{S,T}$, and we will further clarify this in the next version of the paper.

3. **Sensitivity defintion:** We will define sensitivity at an appropriate spot before section 5.2.

[Meta-Review · NeurIPS 2019]

A risk bound for data-dependent hypothesis classes is presented in terms of a notion of stability of the hypothesis class and a newly proposed extension of the Rademacher complexity to data-dependent classes. The paper is clearly written and the results are interesting and mathematically sound. The unification of the complexity-based and stability-based analysis for learning with data-dependent hypothesis seems a significant contribution. Their main theoretical result (Theorem 2) applies to a large range of learning algorithms and is thus relevant to a large body of machine learning work. A nice analysis is presented for bagging, stochastic strongly convex optimisation, and distillation. Hence, we recommend acceptance.